# Inference Time Optimization with Confidence Dynamics

**Yu Wang**[1]  **Minghao Liu**[1]  **Jiayun Wang**[1]  **Jinrui Huang**[1]  **Ankit Shah**[1]  **Wei Wei**[1]

## Abstract

Inference time optimization techniques, such as repeated sampling, have significantly advanced the reasoning capabilities of Large Language Models (LLMs). However, the critical role of model uncertainty remains largely underexplored in these optimization strategies. In this paper, we investigate the dynamics of confidence along reasoning trajectories and for first time reveal a surprising and unique pattern: correct answer traces tend to exhibit confidence improvement over time (positive confidence gain), while incorrect traces show attenuated or declining confidence as reasoning proceeds. Based on this observation, we propose Confidence Dynamic Gain (CDG) based voting, which incorporates how the confidence trajectory of the response evolves along the reasoning chain. Experiments across four open-source architectures (DeepSeek-R1, gpt-oss, Gemma-3, Qwen-QwQ) on the AIME24/25, HMMT25, and BRUMO25 benchmarks demonstrate that CDG yields a significant performance boost over baselines. These results demonstrate that our method provides a robust discriminative signal for improving answer selection in LLM reasoning. We also provide theoretical insights for this phenomenon. Code will be released at https://github.com/Accenture/CDG.git.

## 1. Introduction

Large Language Models (LLMs) have demonstrated remarkable reasoning capabilities across mathematical problem-solving, code generation, and scientific reasoning tasks (DeepSeek-AI, 2025; xAI, 2025; Agarwal et al., 2025; GemmaTeam, 2025; QwenTeam, 2025; Yang et al., 2025). One of the key strategies for improving inference time reasoning accuracy is Best-of-N sampling, where multiple reasoning traces are generated for each given question, and an aggregation strategy selects the final answer (Wang et al., 2022; Snell et al., 2024). The simplest and most widely adopted approach is majority voting (Wang et al., 2022), which selects the answer appearing most frequently among traces.

Despite its simplicity and effectiveness, majority voting treats all traces equally regardless of their quality and uncertainty. Recent state-of-the-art work has explored the incorporation of model confidence into the voting process, using metrics such as sequence-level perplexity or average token probability to weight votes (Kang et al., 2025b; Fu et al., 2025). However, these methods rely on *static* confidence measures that aggregate information across an entire trace, potentially obscuring valuable signals about how the model's certainty evolves within reasoning chains.

In this work, we investigate the *dynamics* of model confidence along the reasoning trajectory. Through extensive experiments across four mainstream LLM architectures—OpenAI gpt-oss (Agarwal et al., 2025), DeepSeek-R1 (DeepSeek-AI, 2025), Google Gemma (GemmaTeam, 2025), and Qwen-QwQ (QwenTeam, 2025)—we uncover a striking empirical pattern: **correct reasoning traces exhibit systematically higher confidence gains from the beginning to the end of the trace compared to incorrect traces**. Specifically, when we partition each trace into position-normalized bins and compare confidence levels between the initial (head) and final (tail) portions, correct traces show positive averaged confidence improvement at the end of the reasoning in comparison to the reasoning start, while incorrect traces display attenuated or negative end-start confidence difference.

This observation leads us to ponder whether such a signal provides additional information to the existing confidence measures for sampling and why. To test our hypothesis, we propose **Confidence Dynamic Gain (CDG) based Voting**. Rather than relying solely on average confidence or answer frequency, CDG explicitly incorporates the confidence dynamic gain —the difference between answer's tail and head confidence—as a discriminative side information for answer selection. Inspired by existing voting method (Fu et al., 2025; Kang et al., 2025b), our answer voting function elegantly integrates three signals: (1) the count of traces for each answer, (2) the mean confidence across traces, and

---

[1]Center for Advanced AI, Accenture. Correspondence to: Yu Wang <feather1014@gmail.com>.

*Proceedings of the 43rd International Conference on Machine Learning*, Seoul, South Korea. PMLR 306, 2026. Copyright 2026 by the author(s).

(3) confidence dynamic gain. Surprisingly, we observe that by simply amplifying such a confidence dynamic signal, CDG achieves strong performance boost compared to the state-of-the-art inference time answer selection approaches.

While we are open to other legitimate explanations, we hypothesize that the training dynamics of Group Relative Policy Optimization (GRPO) (Shao et al., 2024), a well-adopted algorithm to train LLMs, could potentially be the reason behind. GRPO has reinforced tokens proportional to their frequency within the group. In correct traces, while the initial reasoning paths are diverse (low frequency), the final answers strictly converge to the same ground truth (high frequency). In contrast, incorrect traces are subject to negative confidence suppression owing to diluted reasoning paths. With mild assumptions, we show that this **concentration at the tail** for correct traces drives a distinctively sharp rise in confidence—creating the head-to-tail gap that our method exploits at inference time. We hypothesize that the insights here generalize to any advantage-weighted policy gradient with verifiable reward. Our main contributions are:

- We identify an empirical phenomenon: correct reasoning traces exhibit systematically higher confidence gains along the reasoning trajectories than incorrect traces across diverse LLM architectures.

- We propose CDG, a novel voting method that leverages such confidence dynamics for improved answer selection, generalizing existing confidence-based approaches and majority voting.

- We provide theoretical analysis explaining why GRPO-trained models exhibit such discriminative confidence gradients, connecting CDG to the verifiable reward and training policy behind.

- We demonstrate empirical improvements on competitive mathematics benchmarks, showing that CDG outperforms state-of-the-art baselines.

## 2. Related Work

**Inference time scaling.** Current LLMs increasingly succeed by allocating very large amounts of reasoning at inference, a paradigm we call *inference time scaling* (Snell et al., 2024). Two major directions include: a) Increasing reasoning trajectory: previous work like Chain-of-Thought (Wei et al., 2022) proposes to scale reasoning steps by lengthening thinking steps in a single reasoning trajectory. Later models, such as DeepSeek-R1 (Guo et al., 2025), Grok-4 (xAI, 2025) and Qwen3 (Yang et al., 2025), rely on RL to facilitate this form of inference time scaling. b) Best-of-N setting (this paper's focus): Parallel generation is scaled by increasing the number of trajectories and aggregating them. Self-Consistency (Wang et al., 2022) and Best-of-N (Snell

et al., 2024; Kang et al., 2025b) sample multiple candidates and select via voting or a score. Some rely on reward models such as outcome reward models (Cobbe et al., 2021) and process reward models (Uesato et al., 2022). More efficient variants like Dynamic Voting (Xue et al., 2023) and ranked voting (Wang et al., 2025), reduce the required sample count while preserving accuracy. Our proposed CDG aligns with this line of research, aiming to improve inference time accuracy through optimized answer selection.

**Position plays a role in LLM responses**. Transformers do not inherently model order, so it is critical that positional information is injected through positional encodings (e.g., RoPE (Su et al., 2024) and other absolute/relative schemes), which can directly shape attention allocation and, consequently, generation behavior (Su et al., 2024; Shaw et al., 2018). For example, these schemes often lead to position related structural biases like the "Attention Sink" phenomenon (Xiao et al., 2023; Kang et al., 2025a), where massive attention scores are disproportionately allocated to the first token regardless of relevance. Previous work treat these positional artifacts as mechanical features to be managed through architectural improvements (Su & Yuan, 2025; Qiu et al., 2025). Beyond attention-level effects, token position also shapes how models utilize their context: Liu et al. (2024) reports a "lost in the middle" accuracy curve in long-context question answering, with performance degrading sharply when relevant information is placed in the middle of the input, indicating that position systematically biases what LLMs attend to and reproduce. Others also study the position biases in multi-modality settings (Tang et al., 2025; Zhao et al., 2025). Still, the utility of position-dependent signals for inference-time scaling remains underexplored. In this paper, our CDG also leverages position information from the trajectories, exploiting certainty dynamics to guide inference-time scaling for Best-of-N selection.

**Confidence of response traces.** Recent state-of-the-art finds estimating the response confidence during inference improves the model reliability. Various methods estimate model confidence, such as confidence token routing (Chuang et al., 2024) and selective generation (Ren et al., 2023). Kang et al. (2025b) proposes global confidence, which is computed at the sequence level and applied post hoc to rank or select among completed candidates, showing that combining multi-sample reasoning with confidence-aware selection can outperform majority voting while using fewer generated tokens. DeepConf (Fu et al., 2025) introduces a local confidence signal that is updated along each trajectory, triggering on-the-fly pruning of low-confidence traces for more token-efficient parallel generation and higher accuracy. In contrast to these scoring methods that aggregate signals over a full trajectory (Fu et al., 2025; Han et al., 2025), our proposed CDG additionally incorporates dynamic confidence information to improve the final voting.

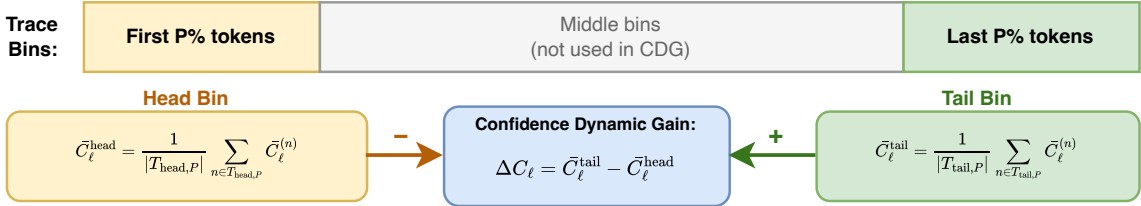

*Figure 1.* Confidence Dynamic Gain (CDG) computation, i.e., $\Delta C_\ell$. Each trace with $T$ tokens is partitioned into $N$ position-normalized bins. The head bin confidence $\bar{C}_\ell^{\text{head}}$ averages over the first $P\%$ of positions, and the tail bin confidence $\bar{C}_\ell^{\text{tail}}$ averages over the last $P\%$. The CDG $\Delta C_\ell$ captures how confidence evolves from reasoning start to conclusion.

## 3. Method

We are motivated to exploit the hidden confidence dynamic in the autoregressive generation trajectory for better LLM reasoning. Through extensive experiments using mainstream open source LLM architecture, we observe an interesting pattern in such confidence dynamics of the output sequences: when we have sampled multiple traces/answers for a single question, the correct traces always tend to have larger "tail-start" confidence gain throughout the reasoning trajectory than wrong traces. To test whether such a phenomenon provides any useful information for sampling, we propose a novel method named Confidence Dynamic Gain (CDG) based voting. We demonstrate that CDG is able to leverage this unique pattern and improve sampling accuracy by suppressing hallucinated answers.

### 3.1. Preliminaries

The standard framework for Large Language Models relies on the Transformer architecture (Vaswani et al., 2017) to generate text autoregressively. The process transforms an input sequence $x = (x_1, \ldots, x_t, \ldots)$ into a target sequence $y = (y_1, \ldots, y_t, \ldots y_T)$ token by token. For any given position $t$ in the output sequence, the model outputs unnormalized $\text{logit}_t \in \mathbb{R}^V$ (where $V = |\mathcal{V}|$ denotes the vocabulary size). These logits are then mapped to a valid probability distribution defined here as $p(y_t|x, y_{<t}) \in [0,1]^V$, which captures the model's estimated likelihood for the next token $y_t$ based on the preceding context.

**Confidence metric based on Kullback-Leibler (KL) Divergence.** By following the definition in (Kang et al., 2025b) we first define the token level confidence metric as the KL divergence between the token distribution and the uniform distribution at the $t$ token position:

$$C_t^* = \text{KL}(U \parallel p(y_t|x, y_{<t})) = \sum_{j=1}^{V} \frac{1}{V} \log\left(\frac{1/V}{p(y_t = j|x, y_{<t})}\right)$$
$$= -\frac{1}{V} \sum_{j=1}^{V} \log\left(V \cdot p(y_t = j|x, y_{<t})\right). \quad (1)$$

Here, $p(y_t = j|x, y_{<t})$ denotes the probability of choosing the $j^{th}$ token in the vocabulary as the output token at the $t$ position of the output sequence. Intuitively, higher model confidence in Eq. (1) translates to larger divergence

between the output distributions and a uniform distribution $U$, hence corresponding to a relatively peakier and informative $p(y_t|x, y_{<t})$ having small uncertainty. On the contrary, a smaller $C_t^*$ signals a flatter distribution with increased uncertainty. Confidence defined as in Eq. (1) is related to but distinct from the forward KL divergence and entropy: Entropy $\propto -\text{KL}(p(y_t|x, y_{<t}) \parallel U)$. By following (Fu et al., 2025), we adopt the approximation of $C_t^*$ by only computing truncated confidence at position $t$ using the top-K probabilities and define the confidence as $C_t$:

$$C_t = -\frac{1}{K} \sum_{j \in \mathcal{K}_t} \log p(y_t = j|x, y_{<t}), \quad (2)$$

where $\mathcal{K}_t$ is the set of size $K < V$ containing the tokens having top-K probabilities for each position $t$. We fix $K = 20$ as in (Fu et al., 2025).

### 3.2. Confidence Trajectories in Repeated Sampling.

Following the set-up in (Wang et al., 2022), we sample $L$ answers/traces from LLM for each question. Our goal is to select the answer with best accuracy. For each reasoning trace $\ell \in [1, L]$, we compute $C_{\ell,t}$ as the confidence of $t^{th}$ token in trace $\ell$ according to equation Eq. (2). We extract a sequence of per-token confidence $\{C_{\ell,1}, C_{\ell,2}, \ldots, C_{\ell,T}\}$ to represent the confidence trajectory of trace $\ell$. We compute two scores for each trace: **(1) trace level confidence**, **(2) trace level Confidence Dynamic Gain (CDG)**. We first follow (Fu et al., 2025) and use the average confidence across all tokens in the trace to represent the trace level confidence of the $\ell^{th}$ trace:

$$\bar{C}_\ell = \frac{1}{T} \sum_{t=1}^{T} C_{\ell,t}. \quad (3)$$

In order to analyze how model confidence evolves across reasoning traces of varying lengths, we propose a position-normalized binning approach. For each trace with $T$ tokens and per-token confidence sequence $\{C_{\ell,1}, C_{\ell,2}, \ldots, C_{\ell,T}\}$, we partition the sequence into $N$ equal-sized bins (default $N = 10$). The bin size is $b = T/N$, and the $n$-th bin $(n \in 1, 2, \ldots, N)$ contains tokens at positions indexed by $t \in [1, T]$:

$$\mathcal{B}_{\ell,n} = \{C_{\ell,t} : \lfloor (n-1) \cdot b \rfloor < t \le \lfloor n \cdot b \rfloor\} \quad (4)$$

where $\lfloor X \rfloor$ applies the floor function to variable $X$ and rounds down to the nearest integer of $X$. The mean confi-

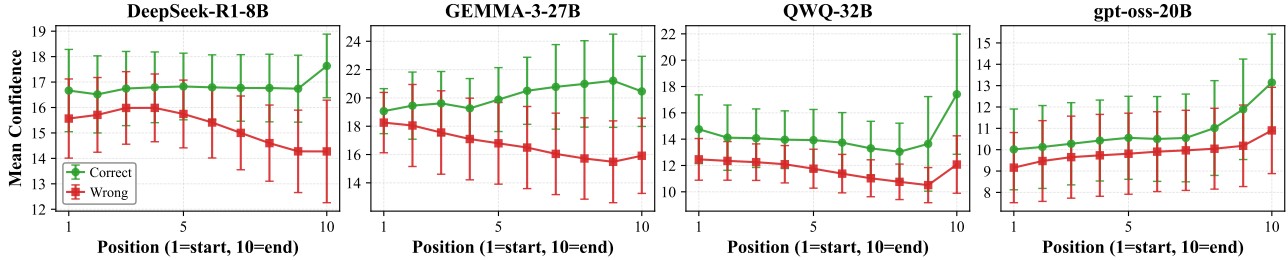

*Figure 2.* Confidence dynamic curves for the reasoning trajectories (partitioned in 10 bins) for different models on AIME 2025 dataset. Significance tests (Table 5 in Appendix) show that correct traces have greater confidence gains than wrong traces.

dence of the $\ell^{th}$ trace for bin $n$ is computed as:

$$\bar{C}_\ell^{(n)} = \frac{1}{|\mathcal{B}_{\ell,n}|} \sum_{C_{\ell,t} \in \mathcal{B}_{\ell,n}} C_{\ell,t}. \tag{5}$$

Here $|\mathcal{B}_{\ell,n}|$ denotes the number of tokens assigned to bin $n$. This normalization maps traces of arbitrary length onto a fixed $N$-dimensional representation $(\bar{C}_\ell^{(1)}, \bar{C}_\ell^{(2)}, \dots, \bar{C}_\ell^{(n)}, \dots \bar{C}_\ell^{(N)})$, where, for instance, bin 1 includes the beginning tokens of reasoning and bin $N$ represents the end. This bin-based confidence statistic has facilitated direct comparison of confidence trajectories along the reasoning chain across traces having different token counts.

In order to observe the confidence trajectories, we evaluate four mainstream LLM architectures: DeepSeek/DeepSeek-R1-8B (DeepSeek-AI, 2025); OpenAI/gpt-oss-20b (Agarwal et al., 2025); Google/gemma-3-27b-it (GemmaTeam, 2025) and Qwen/QwQ-32B (QwenTeam, 2025) on the AIME 2025 benchmark (Balunović et al., 2025). For each model, we sample $L = 512$ reasoning traces per question. Each trace is partitioned into $N = 10$ position-normalized bins, yielding a confidence trajectory $(\bar{C}_\ell^{(1)}, \dots, \bar{C}_\ell^{(10)})$ for each $\ell^{th}$ trace. We aggregate these trajectories by first averaging each bin across $L = 512$ traces within each question, then averaging across all 30 questions. The resulting confidence curves are shown in Figure 2.

Surprisingly, we observe an intriguing pattern in Figure 2: correct and wrong traces exhibit strong distinguishing confidence dynamics patterns: as the reasoning chain proceeds to further token positions till the end, correct traces display obviously greater confidence gains as the reasoning progresses along the reasoning trajectory. In contrast, incorrect traces show attenuated gains or even declining confidence toward the end of the reasoning chain. The error bars indicate standard error across questions, showing that the tail-head difference between correct and incorrect traces is consistent and statistically significant (see Table 5 in Appendix for detailed significance tests).

### 3.3. Confidence Dynamic Gain based Voting

Does such a confidence trajectory evolving pattern provide any useful additional discriminative information to the exist-

ing confidence measures for sampling? To answer this question, we are motivated to propose a novel voting based sampling method called "Confidence Dynamic Gain based Voting", abbreviated as CDG. The CDG method is designed to explicitly investigate whether and how we can leverage the systematic difference in how confidence evolves across the reasoning trajectory to improve answer selection.

Given the $\ell^{th}$ sampled trace with confidence bins $(\bar{C}_\ell^{(1)}, \bar{C}_\ell^{(2)}, \dots, \bar{C}_\ell^{(n)}, \dots \bar{C}_\ell^{(N)})$, we compute the Confidence Dynamic Gain (CDG) as the difference between tail bin and head bin confidence:

$$\Delta C_\ell = \frac{1}{|T_{\text{tail},P}|} \sum_{n \in T_{\text{tail},P}} \bar{C}_\ell^{(n)} - \frac{1}{|T_{\text{head},P}|} \sum_{n \in T_{\text{head},P}} \bar{C}_\ell^{(n)}, \tag{6}$$

where $T_{\text{head},P}$ and $T_{\text{tail},P}$ denote the set having first and last $P\%$ of bins (also tokens) in the $\ell^{th}$ trace. For example, for $N = 10$ total bins, $T_{\text{head},10}$ corresponds to first bin $\mathcal{B}_{\ell,1}$ of the token positions (first 10% token positions), and $T_{\text{tail},10}$ the last 10% token positions.

Intuitively, a **positive** Confidence Dynamic Gain (CDG), i.e., $\Delta C_\ell > 0$ as defined in Eq.(6) indicates the model is gaining *more* confidence as the reasoning chain progresses to final answer (confidence increases), whereas a **negative** $\Delta C_\ell < 0$ indicates the model is losing confidence as the reasoning proceeds to the final conclusion of the answers (confidence decreases). Figure 1 illustrates how $\Delta C_\ell$ is computed and implemented.

Given this observation, it is natural for us to define a trace level quality score for each $\ell^{th}$ trace for voting:

$$s_\ell = \bar{C}_\ell + \beta \cdot \Delta C_\ell, \tag{7}$$

where $\beta$ controls influence of the CDG for voting. It is transparent that in addition to the trace level average confidence $\bar{C}_\ell$, each trace additionally receives an extra credit during the voting: if the term $\Delta C_\ell$ is positive, the score $s_\ell$ is increased, otherwise, the score is penalized for having negative $\Delta C_\ell$. **Selection of $\beta$**. We found the optimal $\beta$ is model specific and scales with $r_b = \mu_C/\Delta_\mu$, where $\mu_C = \mathbb{E}_\ell[\bar{C}_\ell]$ is the expected per-trace mean confidence and $\Delta_\mu = |\mu_{\Delta C}^+ - \mu_{\Delta C}^-|$ is the expected CDG separation between correct and wrong traces using a few

---

**Algorithm 1** Confidence Dynamic Gain based Voting

---

**Require:** Trace answers $\{a_\ell\}_{\ell=1}^L$ and per-token confidence sequences $\{C_{\ell,t}\}_{\ell=1}^L$, hyperparameters $\alpha, \beta, P$
1: **for** each trace $\ell$ **do**
2:    $\bar{C}_\ell = \frac{1}{T}\sum_{t=1}^T C_{\ell,t}$
3:    Compute $\Delta \bar{C}_\ell$ according to Eq. (6)
4:    $s_\ell \leftarrow \bar{C}_\ell + \beta \cdot \Delta C_\ell$
5: **end for**
6: **for** each unique answer $a$ **do**
7:    $\mathcal{T}_a \leftarrow \{\ell : a_\ell = a\}$
8:    $R(a) \leftarrow |\mathcal{T}_a|^\alpha \cdot \mu_a(s_\ell), \forall \ell \in \mathcal{T}_a$
9: **end for**
   **return** $\arg\max_a R(a)$

---

validation questions, where $\mu_{\Delta C}^+ = \mathbb{E}_{\ell \in \text{correct}}[\Delta C_\ell]$ and $\mu_{\Delta C}^- = \mathbb{E}_{\ell \in \text{wrong}}[\Delta C_\ell]$. Empirically, $\beta \in [0.5r_b, 1.5r_b]$ yields robust performance, i.e., $\beta \cdot \Delta C_\ell$ should be comparable with $\bar{C}_\ell$ in magnitude.

**Answer voting**. Inspired by confidence weighting in (Fu et al., 2025), we define final score for each answer $a$ as:

$$R(a) = |\mathcal{T}_a|^\alpha \cdot \mu_a(s_\ell). \quad (8)$$

Here $\mathcal{T}_a$ is the set of all traces producing $a$, and $\mu_a(s_\ell) = \frac{1}{|\mathcal{T}_a|}\sum_{\ell \in a} s_\ell$ calculates the mean value of $s_\ell$ across all $\ell$ traces giving the same answer $a$. The count term $|\mathcal{T}_a|^\alpha$ with range $\alpha \in [0,1]$ controls dampening effect on voting, as we noticed answer count is often dominating (by the effect of majority voting) and weakening the effect of confidence dynamic signal. $\alpha < 1$ weighs less on counting and weighs more on the confidence patterns. It is also worth noting that Eq. (8) generalizes the deepconf (Fu et al., 2025) method and majority vote method, where deepconf is a special case of CDG with $\alpha = 1, \beta = 0$; and majority vote has $\alpha = 1, \mu_a(s_\ell) = 1$ in Eq. (8). Finally, the predicted answer is chosen as the one having the maximum $R(a)$: $\hat{a} = \arg\max_a R(a)$, where $\hat{a}$ is the predicted answer.

## 4. Theoretical Analysis

We aim to explain why confidence trajectories differ between correct and incorrect traces (All proof in Appendix). Our analysis presents simplified assumptions to abstract away architecture details but captures key training dynamics. We expect these insights to extend to other advantage-weighted method beyond GRPO, as the core mechanism — higher tail concentration for correct answers—inherently arises from any training with verifiable rewards.

**Definition 4.1** (GRPO Objective). For input $x$, GRPO samples $G$ answers $\{a_1, \ldots, a_G\}$ with binary rewards $r_i \in \{0, 1\}$. The group-normalized advantage is $A_i = (r_i - \bar{r})/\sigma_r$, where $\bar{r}$ and $\sigma_r$ are the mean and standard deviation of rewards within the group.

**Theorem 4.2** (GRPO Advantage Computation). *In a GRPO batch with $k$ correct and $(G-k)$ incorrect answers ($0 <$*

$k < G$), *the advantages are:*

$$A_{correct} = \sqrt{\frac{G-k}{k}} > 0, \quad A_{incorrect} = -\sqrt{\frac{k}{G-k}} < 0. \quad (9)$$

GRPO assigns uniform positive advantage to correct traces and uniform negative advantage to incorrect traces.

**Assumption 4.3** (Relative Token Concentration). Let $n_t^+(v)$ and $n_t^-(v)$ denote token frequency at position $t$ among correct and incorrect traces generated during training. We assume during training: **(A1) Answer Convergence:** Correct traces converge to ground truth: $n_T^+(v^*) = k$. **(A2) Reasoning Diversity:** With $M \geq 2$ valid approaches, $\max_{v,t<T} n_t^+(v) \leq k/M < k$. **(A3) Relative Concentration Gap:** The tail-to-head ratio for incorrect traces is bounded: $\mathbb{E}[n_T^-]/\mathbb{E}[n_{<T}^-] \leq \gamma M$ for $\gamma < 1$.

Here: (A1) follows from verifiable rewards; (A2) reflects diverse reasoning paths; (A3) Even if the model has a "favorite" wrong answer (a distractor), the probability mass of the incorrect class is inherently fragmented across multiple failure modes, whereas the correct class is usually focused on a singular target during training.

**Assumption 4.4** (Simplified Training Model). Let $\phi_t(v)$ denote the logit of token $v$ at position $t$ and $\Delta\phi_t(v)$ its expected training update. **(M1)** Token logit updates are proportional to advantage-weighted frequency: $\mathbb{E}_{\text{train}}[\Delta\phi_t(v)] \propto A_{\text{correct}} \cdot n_t^+(v) + A_{\text{incorrect}} \cdot n_t^-(v)$; **(M2)** Inference logits approximate base logits plus accumulated training updates; **(M3)** Confidence increases monotonically with selected token logits. In Theorem 4.5, $\Delta\phi_{\text{head}}$ and $\Delta\phi_{\text{tail}}$ denote $\Delta\phi_t(v)$ aggregated over head ($t \ll T$) and tail ($t \approx T$) token positions, respectively. These simplifications abstract away architectural details, with first-order approximations (Assumption Validation in Appendix).

**Theorem 4.5** (Asymmetric Logit Updates). *Under Theorem 4.2 and Assumptions 4.3-4.4, the tail-to-head reinforcement ratios satisfy:*

$$\text{Correct traces:} \quad \frac{\mathbb{E}[\Delta\phi_{tail}]}{\mathbb{E}[\Delta\phi_{head}]} = M \geq 2, \quad (10)$$

$$\text{Incorrect traces:} \quad \frac{\mathbb{E}[\Delta\phi_{tail}]}{\mathbb{E}[\Delta\phi_{head}]} \leq \gamma M \quad where \ \gamma < 1. \quad (11)$$

The key asymmetry: correct traces have tail-to-head ratio $M$, while incorrect traces have lower ratio $\gamma M < M$ (bounded by factor $\gamma < 1$).

**Theorem 4.6** (Confidence Gain Separation). *Under GRPO training with verifiable rewards and Assumptions 4.3-4.4, the expected confidence dynamic gain satisfies:*

$$\mathbb{E}[\Delta C_\ell \mid Correct] - \mathbb{E}[\Delta C_\ell \mid Incorrect] \quad (12)$$

$$\geq c \cdot \eta_{\textit{eff}} \cdot \sqrt{k(G-k)} \cdot \left(\gamma M - \frac{1}{M}\right) > 0, \quad (13)$$

*for constants $c > 0$ and effective learning rate $\eta_{\textit{eff}}$. The separation is positive provided $\gamma > 1/M^2$.*

*Table 1.* Accuracy (%) of different voting methods across models and benchmarks. Best average results are in **bold**. Pass@1 is the single-trace baseline. For CDG (ours), we use $\alpha = 0.5$ and $P\% = 10\%$ across all experiments. $\beta = 10$ for DeepSeek-R1-8B and gpt-oss-20B; $\beta = 3$ for Gemma-3-27B and QWQ-32B. "Majority" is for Majority Vote (Wang et al., 2022). D-CDG is for "Degenerated-CDG", i.e., an ablation of CDG with either $\beta = 0, \alpha = 0.5$ (no confidence gain dynamics, but with count dampening) or $\alpha = 1, \beta \neq 0$ (no count dampening, but with confidence dynamic gain). DC-Mean/DC-Tail are DeepConf-Mean/Tail with top 10% filtering (Fu et al., 2025).

| Model | Dataset | Pass@1 | Majority | DC-Mean | DC-Tail | D-CDG ($\alpha = 1$) | D-CDG ($\beta = 0$) | CDG (ours) |
|---|---|---|---|---|---|---|---|---|
| DeepSeek-R1-8B | AIME 2024 | 86.5 | 90.0 | 90.0 | 93.3 | 93.3 | 90.0 | 93.3 |
| | AIME 2025 | 77.5 | 83.3 | 83.3 | 83.3 | 90.0 | 83.3 | 93.3 |
| | BRUMO 2025 | 79.7 | 93.3 | 93.3 | 93.3 | 93.3 | 93.3 | 93.3 |
| | HMMT 2025 | 59.5 | 70.0 | 70.0 | 83.3 | 76.7 | 70.0 | 83.3 |
| | *Average* | 75.8 | 84.2 | 84.2 | 88.3 | 88.3 | 84.2 | **90.8** |
| Gemma-3-27B | AIME 2024 | 31.5 | 50.0 | 50.0 | 53.3 | 56.7 | 50.0 | 56.7 |
| | AIME 2025 | 24.7 | 30.0 | 30.0 | 46.7 | 33.3 | 30.0 | 40.0 |
| | BRUMO 2025 | 35.9 | 40.0 | 40.0 | 46.7 | 46.7 | 43.3 | 46.7 |
| | HMMT 2025 | 10.8 | 20.0 | 20.0 | 13.3 | 23.3 | 20.0 | 23.3 |
| | *Average* | 25.7 | 35.0 | 35.0 | 40.0 | 40.0 | 35.8 | **41.7** |
| gpt-oss-20B | AIME 2024 | 77.9 | 93.3 | 93.3 | 93.3 | 93.3 | 93.3 | 93.3 |
| | AIME 2025 | 71.1 | 90.0 | 90.0 | 90.0 | 90.0 | 90.0 | 93.3 |
| | BRUMO 2025 | 64.8 | 80.0 | 83.3 | 83.3 | 83.3 | 83.3 | 83.3 |
| | HMMT 2025 | 52.2 | 66.7 | 70.0 | 73.3 | 70.0 | 70.0 | 73.3 |
| | *Average* | 66.5 | 82.5 | 84.2 | 85.0 | 84.2 | 84.2 | **85.8** |
| QwQ-32B | AIME 2024 | 81.7 | 86.7 | 90.0 | 86.7 | 90.0 | 90.0 | 90.0 |
| | AIME 2025 | 71.6 | 76.7 | 76.7 | 76.7 | 76.7 | 76.7 | 76.7 |
| | BRUMO 2025 | 76.8 | 80.0 | 80.0 | 86.7 | 86.7 | 80.0 | 90.0 |
| | HMMT 2025 | 48.7 | 56.7 | 56.7 | 63.3 | 63.3 | 56.7 | 63.3 |
| | *Average* | 69.7 | 75.0 | 75.9 | 78.3 | 79.2 | 75.8 | **80.0** |
| | *Overall Average* | 59.4 | 69.2 | 69.8 | 72.9 | 72.9 | 70.0 | **74.6** |

**Interpretation.** The core mechanism here leading to Theorem 4.6 is the asymmetry in signal concentration. For *correct traces*, diverse reasoning paths tend to collapse into the same answer, creating a high tail-to-head concentration ratio $M$. This amplifies the positive reinforcement at the tail, driving a sharp confidence increase. For *incorrect traces*, under the assumption that the concentration ratio is lower ($\gamma M < M$) than correct traces because during training answer errors are inherently more diverse than the unique ground truth, we are able to show that the positive "confidence spike" for correct answers significantly outpaces any concentration in incorrect traces, creating the systematic separation gap exploited by CDG.

## 5. Empirical Results

We conduct comprehensive experiments to evaluate the effectiveness of CDG across multiple dimensions: (1) comparison against baseline methods on challenging mathematical reasoning benchmarks, (2) ablation studies, and (3) analysis of score distributions of different methods. We benchmark CDG against state-of-the-art voting baselines, including majority voting (Wang et al., 2022), DeepConf-Mean, and DeepConf-Tail (Fu et al., 2025), across four challenging mathematics datasets. More complete results and analysis

are included in Appendix.

### 5.1. Experimental Setup

**Models.** We evaluate four state-of-the-art open-source reasoning LLMs, representing diverse design choices and training methodologies: (1) **DeepSeek/DeepSeek-R1-8B** (DeepSeek-AI, 2025); (2) **OpenAI/gpt-oss-20B** (Agarwal et al., 2025); (3) **Google/gemma-3-27b-it** (GemmaTeam, 2025), an open-weight model; and (4) **Qwen/QwQ-32B** (QwenTeam, 2025). This diverse selection enables us to assess the generalizability of confidence dynamics patterns across different model families. For **gpt-oss-20B**, we disabled the "high reason" setting; this allows us to more clearly observe the sampling dynamics given the model's imperfect reasoning capacity. We excluded Qwen-3 model (Yang et al., 2025) as its post-RL training involves entropy control, mechanisms that directly disrupt the confidence trajectory patterns and thus violate the assumptions underlying the theorems studied in this work.

**Benchmarks.** Following the evaluation protocol of state-of-the-art answer selection methods (Fu et al., 2025), we compare different methods on following benchmarks: (1) **AIME 2024** (Balunović et al., 2025) (American Invitational Mathematics Examination); (2) **AIME 2025** (Balunović

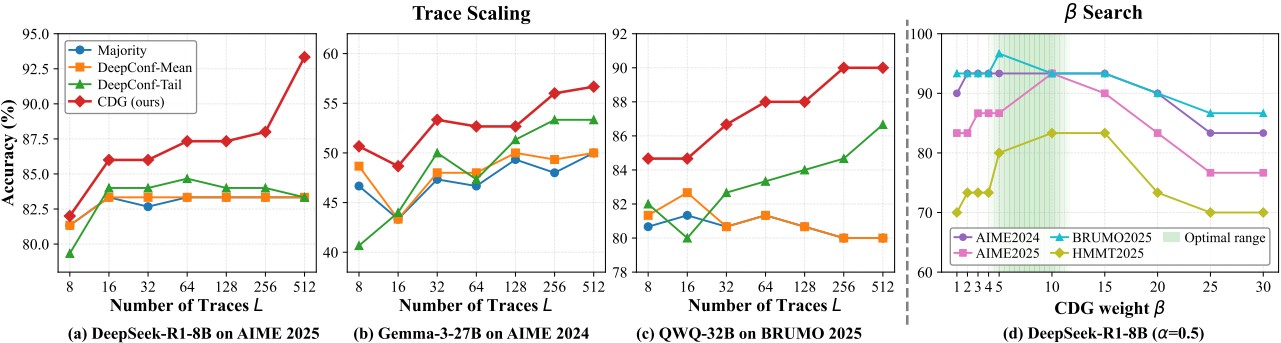

*Figure 3.* Ablation study of trace number (budget) $L$ and CDG weight $\beta$. (a) Accuracy vs $L$ for DeepSeek-R1-8B on AIME 2025; (b) Accuracy vs $L$ for Gemma-3-27B on AIME 2024. (c) Accuracy vs $L$ for QwQ-32B on BRUMO 2025. (d)Accuracy vs $\beta$ for CDG using DeepSeek-R1-8B across 4 datasets. The green band highlights the estimated optimal range of $\beta \in [0.5r_b, 1.5r_b]$ as discussed in Section 3.3. See Figures 5 and 6 in the Appendix for complete results.

et al., 2025); (3) **HMMT 2025** (HMMT, 2025)(Harvard-MIT Mathematics Tournament); and (4) **BRUMO 2025** (Brumo, 2025) (Bulgarian Mathematical Olympiad). These benchmarks have multi-step reasoning trajectories and verifiable ground truth, which is ideal for evaluating CDG and justification of the theories. We also include partial results on GPQA-Diamond (Rein et al., 2024) with DeepSeek-R1-8B in the Appendix (Table 8), to show the generalizability of CDG beyond mathematical reasoning.

**Baselines.** We compare CDG against three state-of-art voting methods: **(1) Majority Voting (Self-Consistency)** (Wang et al., 2022): Selects the answer with the highest count among sampled traces. **(2) DeepConf-Mean** (Fu et al., 2025): Weights answer votes by the average trace-level confidence $\bar{C}_\ell = \frac{1}{T}\sum_{t=1}^{T} C_{\ell,t}$. **(3) DeepConf-Tail** (Fu et al., 2025): Filter to top $10\%$ traces by tail conf (2048 tail tokens only), then use weighted vote by the confidence of 2048 tail tokens only. **(4) Pass@1** results as lower bound. We implement all baselines using official published code.

**Implementation details.** For each question, we sample $L = 512$ reasoning traces. Each trace is partitioned into $N = 10$ position-normalized bins using the procedure in Section 3.2. For CDG, we use default hyperparameters $\alpha = 0.5$ (count dampening) and $P\% = 10\%$ (head/tail percentile) across all the experiments. We set $\beta = 10$ for DeepSeek-R1-8B and gpt-oss-20B; $\beta = 3$ for Gemma-3-27B and QwQ-32B, based on the $\beta$ selection logic in Section 3.3. Inference time parameters are in Appendix (Table 9). To estimate $\beta$, we used a rotating cross-benchmark calibration scheme: $\beta$ is estimated on one benchmark (e.g., AIME 2024) and applied to the other three (AIME 2025, HMMT 2025, BRUMO 2025); we repeated this with each dataset serving in turn as the calibration set.

### 5.2. Main Results

Table 1 presents the accuracy of different answer selection methods. "Majority" is Majority Vote (Self-consistency)

(Wang et al., 2022). "DC-Mean" is DeepConf-Mean, "DC-Tail" is DeepConf-Tail with top $10\%$ filtering as in (Fu et al., 2025). "D-CDG" is for "Degenerated-CDG", i.e., an ablation of CDG (see Section 5.3).

**CDG outperforms baselines.** CDG achieves the highest averaged accuracy across all models, demonstrating the consistent effectiveness of incorporating confidence dynamics into voting. On the overall average across all models and datasets, CDG improves over majority voting by $5.4\%$ and over DeepConf-Mean by $4.8\%$. For DeepConf-Tail method which uses a fixed set of filtered tail 2048 tokens for weighted voting, CDG without any filtering also is able to get an $1.7\%$ improvement. Most importantly, we show in section **Number of Traces** and Figure 3 that CDG maintains its superiority across varying $L$. Statistical significance test of CDG improvement over baseline methods is also presented in Table 6 Appendix, showing reliability of CDG. The improvements are particularly pronounced on harder benchmarks (AIME 2025), where the discriminative signal from confidence dynamics is more valuable. DeepConf-Mean and DeepConf-Tail both capture useful confidence information, but ignoring the evolving dynamics in traces. In comparison, CDG has explicitly captured this contrast signal $\Delta C_\ell$, leveraging the theoretical insight in Section 4.

### 5.3. Ablation Study

To ablate the effect of $\alpha$ and $\beta$, we include "Degenerated-CDG" (abbreviated as D-CDG) in Table 1, i.e., an ablation of CDG with either $\beta = 0, \alpha = 0.5$ (no confidence gain dynamics, but with count dampening) or $\alpha = 1, \beta \neq 0$ (no count dampening, but with confidence dynamic gain). We include an ablation on $P$ in the Appendix (Figure 7). All results reported in this subsection have variance below $0.09\%$ across random-seed runs.

**Count dampening weight $\alpha$.** The count dampening term $|\mathcal{T}_a|^\alpha$ (where $\alpha < 1$) mitigates the dominance of majority voting over the confidence signal. Without dampening ($\alpha =$

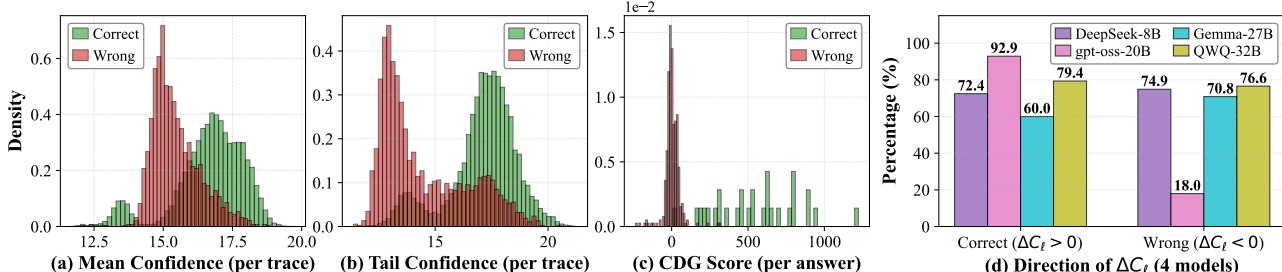

*Figure 4.* Confidence metric statistical analysis on AIME 2024 using DeepSeek-R1-8B. Wrong traces are marked in red while correct ones in green.(a) Mean token confidence distribution. (b) Tail token confidence (last 10%). (c) CDG score (d) Direction analysis for $\Delta C_\ell$.

1), raw answer counts can overwhelm subtle variations in confidence, effectively erasing the benefits of CDG. As shown in Table 1, the Degenerated-CDG (D-CDG) baseline with $\alpha = 1, \beta \neq 0$ performs significantly worse than the complete CDG, verifying that count dampening is a critical component for the efficacy of CDG.

**CDG weight $\beta$.** We study the effect of $\beta$, which controls the weight of CDG relative to mean confidence. In Table 1, Degenerated-CDG (D-CDG) $\beta = 0, \alpha = 0.5$ completely removes the effect of $\Delta C_\ell$, and ignores trajectory dynamics, which only has the effect of Count dampening with $\alpha = 0.5$. D-CDG therefore shows similar performance to DeepConf-Mean according to the definition in Eq. (8). Also, Figure 3(d) shows accuracy across different $\beta$ values on AIME 2024. When $\beta$ decreases, CDG weighs less on trajectory dynamics, and gradually reduces to the performance of DeepConf-Mean. As $\beta$ increases, CDG increasingly favors traces with positive confidence gains peaking at $\beta \approx 10$. Beyond the estimated optimal range (green band), performance plateaus or slightly decreases due to over-reliance on the dynamic signal and ignoring the overall trace confidence. This confirms that CDG provides complementary information to static confidence.

**Number of traces $L$ (Budget).** For each question, we subsample $L \in \{8, 16, 32, 64, 128, 256\}$ traces from our full 512 sample pool, shuffle the traces, and sample 5 random runs for each $L$, we then evaluate voting accuracy against varying $L$. Figure 3(a)(b)(c) show the accuracy curves for CDG, majority voting, and DeepConf methods. While all methods improve with more samples, CDG exhibits a steeper improvement curve than other state-of-the-arts, and maintains consistent advantage across all sample sizes $L$ with the same set of hyperparameters as used in Table 1. This ablation reveals CDG's advantage persists across different computational budgets and confidence dynamics provide consistent benefits regardless of $L$. Full results of this ablation study are in Appendix.

**CDG is not merely reinforced tail confidence.** We also ablate the effect of CDG by comparing full CDG , i.e., $\Delta C_\ell$ in Eq. (6) against using only tail confidence with-

*Table 2.* Ablation on $\Delta C_\ell$. Metric: Accuracy (%). CDG uses full $\Delta C_\ell$ in Eq. (6), "No Start" uses only tail confidence for $\Delta C_\ell$ without subtraction. All use same parameters as in Table 1. Drop is performance drop. Full table in Appendix.

| Model | CDG | No Start | Drop |
|---|---|---|---|
| DeepSeek-R1-8B | **90.8** | 84.2 | -6.6 |
| Gemma-3-27B | **41.7** | 36.7 | -5.0 |
| gpt-oss-20B | **85.8** | 84.2 | -1.6 |
| QwQ-32B | **80.0** | 75.8 | -4.2 |
| *Overall Average* | ***74.6*** | *70.2* | *-4.4* |

out subtraction of start confidence ("No Start" in Table 2), i.e., $\Delta \tilde{C}_\ell = \frac{1}{|T_{\text{tail}, P}|} \sum_{n \in T_{\text{tail}, P}} \bar{C}_\ell^{(n)}$. Results in Table 2 demonstrate that the CDG component is essential: removing the start confidence subtraction decreases accuracy, with losses even up to $6.6\%$ on DeepSeek-R1 (and $13.3\%$ drop on HMMT25, see Table 3 in Appendix). This confirms that the CDG captures reasoning dynamics rather than just leveraging on reinforced tail confidence in the voting: the start confidence subtraction normalizes for initial confidence levels and rewards traces showing confidence growth during reasoning. This also explains CDG's evident performance boost over DeepConf-Tail in Table 1.

**Trace length**. We also investigate the effect of trace length on CDG performance. For each question, we split 512 traces into a short pool (256 traces) and a long pool (256 traces), then ran CDG on each pool. Note in this case, the trace distribution therefore changed from the original answer distribution with 512 traces. We found CDG maintains advantage in both pools regardless of controlled lengths, We also found in general correct traces are shorter (t-test, $p < 0.001$), but Gemma on HMMT shows where correct traces are longer. These results suggest that CDG's advantage is not dependent on trace length differences, but rather the confidence dynamics along the reasoning trajectory. See Table 7 in Appendix B.6 for detailed number comparisons.

**Effect of `\boxed{}` tokens.** Since the final answer in all four benchmarks is extracted from the `\boxed{}` token, a natural concern is whether CDG's signal is merely an artifact of these high-confidence formatting tokens at the trace tail.

To rule this out, we recompute CDG across all 16 model–dataset configurations after excluding \boxed{} tokens from the confidence trajectory. The resulting answer selections are essentially unchanged: the average accuracy difference is 0.0%, with 99% per-question selection agreement (the rare disagreements pick different traces that map to the same final answer). Intuitively, \boxed{} formatting carries large confidence in every trace and therefore cancels in the head-vs-tail comparison underlying CDG. This indicates that CDG's advantage is not driven by answer-formatting tokens, but by the genuine evolution of confidence along the reasoning trajectory.

### 5.4. Score Distribution Analysis

To understand why CDG achieves superior voting accuracy, we analyze the distributions of different scoring metrics for correct versus incorrect traces. For each trace in our dataset, we compute three scores: (1) **Mean Confidence** $\bar{C}_\ell$: the average confidence across all tokens (used by DeepConf-Mean); (2) **Tail Confidence**: the mean confidence of tail 2048 tokens (used by DeepConf-Tail); and (3) **CDG Score** as in Eq. (8): our proposed score combining mean confidence and confidence dynamic gain and the effect of counts. We visualize these distributions as histograms, separately for correct (green) and incorrect (red) traces. Figure 4 presents the score distributions for DeepSeek-R1 on AIME 2024. Among all scoring metrics, CDG exhibits the clearest separation between correct and incorrect trace distributions. The histograms show minimal overlap between the two groups, indicating that CDG score is highly discriminative. We also study the population statistics of dynamic gain direction across four models in Figure 4(d), revealing that most of correct traces exhibit positive confidence gradients ($\Delta C_\ell > 0$) while majority of wrong traces show negative gradients ($\Delta C_\ell < 0$). This pattern confirms that reasoning quality correlates with increasing confidence trajectories across the population, providing empirical justification for CDG's motivation. For Gemma on AIME24, although we do not observe a significant frequency of negative CDG values for wrong traces, we observe attenuated gains compared to correct traces (Figure 2), which also explains the performance boost of CDG using Gemma model over baselines. Statistical significance tests for CDG's improvement over baselines are in Appendix (Table 6).

## 6. Conclusion

We reveal a novel discriminative pattern in LLM responses: correct traces exhibit systematically higher confidence gains from beginning to end compared to incorrect traces across diverse model architectures. Building on this insight, we proposed Confidence Dynamic Gain (CDG) voting, which incorporates confidence trajectory information into answer selection and generalizes majority voting and static con-

fidence methods as special cases. We provide theoretical justification through analysis of GRPO training dynamics, showing that asymmetric advantage assignment combined with maximal token concentration at answer positions naturally induces divergent confidence gradients. Experiments on challenging benchmarks demonstrate consistent improvements over baselines, establishing CDG as a principled training-free answer selection method in LLM reasoning.

## Impact Statement

This paper presents work whose goal is to advance the field of Machine Learning, specifically improving answer selection in LLM reasoning through confidence dynamics analysis. Our method is training-free and operates on existing model outputs, introducing negligible additional computational or environmental costs beyond standard inference. By improving the reliability of LLM reasoning without requiring model retraining, CDG voting could contribute to safer deployment of AI systems in educational and scientific applications. We do not foresee specific negative societal consequences that must be highlighted here. We also note several limitations of the current work that scope these claims. **Compute and latency of Best-of-$N$ inference.** CDG inherits the cost profile of repeated sampling: generating $L$ traces per query multiplies both compute and wall-clock latency relative to single-sample decoding. The CDG aggregation itself is essentially free (a single pass over per-token logprobs), but trace generation dominates the cost. Our scaling analysis (Figure 3 and Figure 5 in Appendix) shows CDG retains its advantage even at small $L$, partially mitigating this overhead, but Best-of-$N$ remains less suitable for strict latency-bound deployments. **Benchmark scope.** Our main evaluation focuses on competition-level mathematical reasoning (AIME 2024/2025, HMMT 2025, BRUMO 2025), with an initial generalization study on GPQA-Diamond (Appendix B.7). A broader evaluation on code generation (e.g., LiveCodeBench) and general reasoning (e.g., MMLU-Pro, BBH) is left to future work to more fully characterize the scope of confidence dynamics across task types. **Token-level confidence access and model-specific tuning.** CDG requires top-$K$ token logprobs at every decoding step, which is currently exposed by open-source serving stacks (e.g., vLLM) but not always by all closed-source APIs. In addition, the CDG weight $\beta$ is model-dependent; we provide a principled scaling rule $\beta \in [0.5r_b, 1.5r_b]$ (Section 3.3), but estimating $r_b$ still requires a small per-model validation set rather than being fully automatic.

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

# A. Proofs

This section of the appendix provides detailed proofs for all theoretical results in Section 4, empirical validation of assumptions, and supplementary discussion.

## A.1. Proof of Theorem 4.2 (GRPO Advantage Computation)

*Proof.* Consider a GRPO training batch with $G$ answers, where $k$ are correct ($r_i = 1$) and $(G - k)$ are incorrect ($r_i = 0$).

**Step 1: Mean reward.** $\bar{r} = \frac{1}{G} \sum_{j=1}^{G} r_j = \frac{k}{G}$.

**Step 2: Variance.** Using $\sigma^2 = \mathbb{E}[r^2] - (\mathbb{E}[r])^2$:

$$\mathbb{E}[r^2] = \frac{k}{G}, \quad (\mathbb{E}[r])^2 = \frac{k^2}{G^2}, \quad \Rightarrow \quad \sigma_r^2 = \frac{k(G-k)}{G^2}, \quad \sigma_r = \frac{\sqrt{k(G-k)}}{G}. \tag{14}$$

**Step 3: Advantages.**

$$A_{\text{correct}} = \frac{1 - k/G}{\frac{\sqrt{k(G-k)}}{G}} = \frac{G - k}{\sqrt{k(G-k)}} = \sqrt{\frac{G - k}{k}} > 0, \tag{15}$$

$$A_{\text{incorrect}} = \frac{0 - k/G}{\frac{\sqrt{k(G-k)}}{G}} = \frac{-k}{\sqrt{k(G-k)}} = -\sqrt{\frac{k}{G - k}} < 0. \tag{16}$$

$\square$

## A.2. Discussion of Assumption 4.4 (Simplified Training Model)

Assumption 4.4 introduces simplifications to make analysis tractable:

**(M1) Token-Level Gradient Aggregation.** We model cumulative training effect on logit $\phi_t(v)$ as proportional to advantage-weighted frequency:

$$\mathbb{E}_{\text{train}}[\Delta\phi_t(v)] \propto A_{\text{correct}} \cdot n_t^+(v) + A_{\text{incorrect}} \cdot n_t^-(v). \tag{17}$$

This treats logits as if they were direct parameters, whereas in reality $\phi_t = W_{\text{out}} h_t$ depends on all transformer layers. The approximation is valid to the extent that gradients $\frac{\partial \mathcal{L}}{\partial \phi_t}$ accumulate proportionally to advantage weights, ignoring attention coupling and layer-wise gradient flow. We give the proof and validation of this assumption as follows.

**Formal Derivation of Token-Level Gradient Aggregation (M1)**

We show that (M1) follows directly from the GRPO policy gradient structure.

**Proposition A.1** (Token-Level Gradient Aggregation)**.** *Under the GRPO objective, the cumulative gradient update for token $v$ at position $t$ satisfies:*

$$\Delta \log \pi_\theta(v \mid x_{<t}) \propto A_{\text{correct}} \cdot n_t^+(v) + A_{\text{incorrect}} \cdot n_t^-(v). \tag{18}$$

*Proof.* **Step 1: GRPO gradient.** The GRPO policy gradient aggregates over all $G$ traces:

$$\nabla_\theta \mathcal{L} = \sum_{i=1}^{G} A_i \cdot \nabla_\theta \log \pi_\theta(a_i \mid x). \tag{19}$$

**Step 2: Token-level decomposition.** For a specific token $v$ at position $t$, the gradient contribution comes from the subset of traces that generate $v$ at position $t$. Let $\mathcal{T}_v = \{i : a_i^t = v\}$. Then:

$$\Delta \log \pi_\theta(v \mid x_{<t}) \propto \sum_{i \in \mathcal{T}_v} A_i. \tag{20}$$

**Step 3: Partition by correctness.** Since all correct traces share $A_{\text{correct}}$ and all incorrect traces share $A_{\text{incorrect}}$ (Theorem 4.2):

$$\sum_{i \in \mathcal{T}_v} A_i = A_{\text{correct}} \cdot \underbrace{|\mathcal{T}_v \cap \text{Correct}|}_{n_t^+(v)} + A_{\text{incorrect}} \cdot \underbrace{|\mathcal{T}_v \cap \text{Incorrect}|}_{n_t^-(v)}. \tag{21}$$

$\square$

The result is a direct consequence of the GRPO gradient structure: each trace contributes its advantage $A_i$ to the update of every token it contains. Since advantages are constant within correctness groups (Theorem 4.2), the total update factors into advantage-weighted frequency counts. This justifies (M1) without additional assumptions beyond the GRPO objective itself.

**(M2) First-Order Inference Approximation.** We model:

$$\phi_{\text{inference}}(v) \approx \phi_{\text{base}}(v) + \eta_{\text{eff}} \cdot \mathbb{E}_{\text{train}}[\Delta\phi(v)], \tag{22}$$

where $\eta_{\text{eff}}$ aggregates training effects. This ignores non-linearity, normalization layers, distributional shift, and gradient evolution over epochs. Despite these limitations, it captures that training systematically shifts logits along advantage-weighted gradients.

**(M3) Confidence Monotonicity.** Confidence $C_t$ (KL divergence from uniform) increases with selected token logit. Formalized in Lemma A.2 below.

**Summary.** These provide a simplified toy model explaining qualitative patterns rather than quantitative predictions.

**Proof of Lemma: Confidence-Logit Relationship**

**Lemma A.2** (Confidence-Logit Relationship). *For confidence $C_t = -\frac{1}{K}\sum_{j \in top\text{-}K} \log p_j$ where $p_j = \frac{e^{\phi_j}}{\sum_k e^{\phi_k}}$, if the selected token $v_t$ is among top-K and its logit increases by $\Delta\phi_t(v_t) > 0$ while others remain constant:*

$$\Delta C_t \geq \frac{1}{K} \cdot \frac{\Delta\phi_t(v_t)}{1 + e^{\Delta\phi_t(v_t)}} > 0. \tag{23}$$

*Proof.* Let $C(\phi) = -\frac{1}{K}\sum_{j \in \mathcal{K}} \log p_j(\phi)$ where $\mathcal{K}$ is the top-K set.

**Step 1: Compute derivative.** For $i \in \mathcal{K}$:

$$\frac{\partial \log p_i}{\partial \phi_i} = \frac{\partial}{\partial \phi_i}\left(\phi_i - \log \sum_k e^{\phi_k}\right) = 1 - p_i, \tag{24}$$

$$\frac{\partial \log p_j}{\partial \phi_i} = -p_i \quad \text{for } j \neq i. \tag{25}$$

Thus:

$$\frac{\partial C}{\partial \phi_i} = -\frac{1}{K}\sum_{j \in \mathcal{K}} \frac{\partial \log p_j}{\partial \phi_i} = -\frac{1}{K}\left[(1 - p_i) - (K-1)p_i\right] = \frac{1}{K}(Kp_i - 1). \tag{26}$$

For selected token $v_t$ in top-K, typically $p_{v_t} > 1/K$, giving $\frac{\partial C}{\partial \phi_{v_t}} > 0$.

**Step 2: Lower bound via probability analysis.** We analyze the change in confidence when $\phi_i$ increases, treating the top-K set $\mathcal{K}$ as fixed (a valid first-order approximation when the selected token is already in top-K and perturbations are moderate).

When $\phi_i$ increases by $\Delta\phi_i$, the partition function changes to $Z' = Z + e^{\phi_i}(e^{\Delta\phi_i} - 1) = Z(1 + p_i(e^{\Delta\phi_i} - 1))$.

The new probabilities are:

$$p_i' = \frac{e^{\phi_i + \Delta\phi_i}}{Z'} = \frac{p_i e^{\Delta\phi_i}}{1 + p_i(e^{\Delta\phi_i} - 1)}, \tag{27}$$

$$p_j' = \frac{e^{\phi_j}}{Z'} = \frac{p_j}{1 + p_i(e^{\Delta\phi_i} - 1)} \quad \text{for } j \neq i. \tag{28}$$

The confidence change over the fixed top-K set is:

$$\Delta C = -\frac{1}{K} \sum_{j \in \mathcal{K}} (\log p'_j - \log p_j) \tag{29}$$

$$= -\frac{1}{K} \left[ \Delta\phi_i - \log(1 + p_i(e^{\Delta\phi_i} - 1)) - (K-1)\log(1 + p_i(e^{\Delta\phi_i} - 1)) \right] \tag{30}$$

$$= \frac{1}{K} \left[ K\log(1 + p_i(e^{\Delta\phi_i} - 1)) - \Delta\phi_i \right]. \tag{31}$$

For the selected token $v_t$ with $p_{v_t} \geq 1/K$, let $x = e^{\Delta\phi_{v_t}} - 1 \geq 0$. Then:

$$\Delta C = \frac{1}{K}[K\log(1 + p_{v_t}x) - \log(1 + x)]. \tag{32}$$

Since $p_{v_t} \geq 1/K$, we have $p_{v_t}x \geq x/K$. Using the inequality $\log(1+y) \geq \frac{y}{1+y}$ for $y \geq 0$ and noting that $e^{\Delta\phi_{v_t}} - 1 \geq \frac{\Delta\phi_{v_t}}{e^{\Delta\phi_{v_t}}}$ (which follows from $e^y(1 - y/e^y) \leq 1$ for $y \geq 0$):

$$\Delta C \geq \frac{1}{K} \cdot \frac{\Delta\phi_{v_t}}{1 + e^{\Delta\phi_{v_t}}} > 0. \tag{33}$$

$\square$

## A.3. Proof of Theorem 4.5 (Asymmetric Logit Updates)

*Proof.* By Assumption 4.4(M1):

$$\mathbb{E}_{\text{train}}[\Delta\phi_t(v)] \propto A_{\text{correct}} \cdot n_t^+(v) + A_{\text{incorrect}} \cdot n_t^-(v). \tag{34}$$

**Correct traces.** By Assumption 4.3(A1), $n_T^+(v^*) = k$ (all converge to ground truth). By (A2), $\max_{t<T} n_t^+(v) \leq k/M$ (diverse reasoning). Since $n_T^-(v^*) = 0$ (incorrect traces produce different answers):

$$\mathbb{E}[\Delta\phi_{\text{tail}}^{\text{correct}}] \propto A_{\text{correct}} \cdot k = \sqrt{k(G-k)}, \tag{35}$$

$$\mathbb{E}[\Delta\phi_{\text{head}}^{\text{correct}}] \propto A_{\text{correct}} \cdot \frac{k}{M} = \frac{1}{M}\sqrt{k(G-k)}. \tag{36}$$

Therefore, the ratio is:

$$\frac{\mathbb{E}[\Delta\phi_{\text{tail}}^{\text{correct}}]}{\mathbb{E}[\Delta\phi_{\text{head}}^{\text{correct}}]} = \frac{k}{k/M} = M. \tag{37}$$

**Incorrect traces.** By the generic form:

$$\mathbb{E}[\Delta\phi_{\text{tail}}^{\text{incorrect}}] \propto A_{\text{incorrect}} \cdot \mathbb{E}[n_T^-], \tag{38}$$

$$\mathbb{E}[\Delta\phi_{\text{head}}^{\text{incorrect}}] \propto A_{\text{incorrect}} \cdot \mathbb{E}[n_{<T}^-]. \tag{39}$$

Therefore, the ratio is:

$$\frac{\mathbb{E}[\Delta\phi_{\text{tail}}^{\text{incorrect}}]}{\mathbb{E}[\Delta\phi_{\text{head}}^{\text{incorrect}}]} = \frac{\mathbb{E}[n_T^-]}{\mathbb{E}[n_{<T}^-]} \leq \gamma M \quad \text{by Assumption 4.3(A3).} \tag{40}$$

$\square$

## A.4. Proof of Theorem 4.6 (Confidence Gain Separation)

*Proof.* By Assumption 4.4(M2), $\phi_{\text{inference}} = \phi_{\text{base}} + \eta_{\text{eff}} \cdot \mathbb{E}_{\text{train}}[\Delta\phi]$.

**Step 1: Correct traces.** Confidence gain $\Delta C^{\text{correct}} = C_{\text{tail}} - C_{\text{head}}$. By Lemma A.2 and Theorem 4.5:

$$\mathbb{E}[\Delta C^{\text{correct}}] \gtrsim \frac{\eta_{\text{eff}}}{K} \left[ \mathbb{E}[\Delta\phi_{\text{tail}}] - \mathbb{E}[\Delta\phi_{\text{head}}] \right] = \frac{\eta_{\text{eff}} \cdot c_1}{K} \cdot \sqrt{k(G-k)} \left( 1 - \frac{1}{M} \right), \tag{41}$$

where $c_1 > 0$ is the proportionality constant.

**Step 2: Incorrect traces.** Updates are negative (suppression). Tail receives stronger suppression:

$$\mathbb{E}[\Delta C^{\text{incorrect}}] \lesssim \frac{\eta_{\text{eff}}}{K} \cdot c_1 \cdot A_{\text{incorrect}} \cdot \mathbb{E}[n^-_{<T}] \left[ \frac{\mathbb{E}[n^-_T]}{\mathbb{E}[n^-_{<T}]} - 1 \right] \tag{42}$$

$$\leq \frac{\eta_{\text{eff}}}{K} \cdot c_1 \cdot A_{\text{incorrect}} \cdot \mathbb{E}[n^-_{<T}] \cdot (\gamma M - 1). \tag{43}$$

Since $A_{\text{incorrect}} < 0$ and assuming $\mathbb{E}[n^-_{<T}] \sim \mathcal{O}(\sqrt{k(G-k)})$:

$$\mathbb{E}[\Delta C^{\text{incorrect}}] \lesssim -\frac{\eta_{\text{eff}}}{K} \cdot c_1 \cdot \sqrt{k(G-k)} \cdot (\gamma M - 1). \tag{44}$$

**Step 3: Separation.**

$$\mathbb{E}[\Delta C \mid \text{Correct}] - \mathbb{E}[\Delta C \mid \text{Incorrect}] \tag{45}$$

$$\geq \frac{\eta_{\text{eff}} \cdot c_1}{K} \cdot \sqrt{k(G-k)} \left[ \left(1 - \frac{1}{M}\right) + (\gamma M - 1) \right] = \frac{\eta_{\text{eff}} \cdot c_1}{K} \cdot \sqrt{k(G-k)} \left( \gamma M - \frac{1}{M} \right). \tag{46}$$

For $M \geq 2$ and $\gamma > 1/(M^2)$, this is strictly positive. Setting $c = c_1(\gamma M - 1/M)/K$ gives the result. $\qquad\square$

## A.5. Generalization Beyond GRPO

While we focused on GRPO (Theorem 4.2), we expect these insights to extend to other advantage-weighted policy gradient methods where: (1) positive advantages reinforce correct traces proportionally to token frequency, (2) negative advantages suppress incorrect traces, and (3) token frequencies exhibit the concentration pattern (Assumption 4.3). This likely includes PPO variants with group normalization, reward-weighted regression, and similar RL algorithms with verifiable rewards. While the specific advantage form may differ, the qualitative mechanism (frequency-dependent reinforcement creating head-tail gaps) should remain if these structural properties hold.

# B. More empirical results

## B.1. Complete result for Table 2 in main paper

Table 3 in this Appendix presents the full results summarized as in Table 2 in the main paper.

*Table 3.* Ablation study on gradient differential. CDG uses full gradient (tail−start), "No Start" uses only tail. All use $\alpha = 0.5$, $P = 10\%$. DeepSeek/gpt-oss: $\beta = 10$; Gemma/QwQ: $\beta = 3$.

| Model | Dataset | CDG | No Start | Drop |
|---|---|---|---|---|
| DeepSeek | AIME'24 | **93.3** | 90.0 | -3.3 |
| | AIME'25 | **93.3** | 83.3 | -10.0 |
| | BRUMO'25 | **93.3** | 93.3 | 0.0 |
| | HMMT'25 | **83.3** | 70.0 | -13.3 |
| | *Avg* | *90.8* | *84.2* | *-6.6* |
| Gemma | AIME'24 | **56.7** | 50.0 | -6.7 |
| | AIME'25 | **40.0** | 30.0 | -10.0 |
| | BRUMO'25 | **46.7** | 46.7 | 0.0 |
| | HMMT'25 | **23.3** | 20.0 | -3.3 |
| | *Avg* | *41.7* | *36.7* | *-5.0* |
| QwQ | AIME'24 | **90.0** | 90.0 | 0.0 |
| | AIME'25 | **76.7** | 76.7 | 0.0 |
| | BRUMO'25 | **90.0** | 80.0 | -10.0 |
| | HMMT'25 | **63.3** | 56.7 | -6.6 |
| | *Avg* | *80.0* | *75.8* | *-4.2* |
| gpt-oss | AIME'24 | **93.3** | 93.3 | 0.0 |
| | AIME'25 | **93.3** | 90.0 | -3.3 |
| | BRUMO'25 | **83.3** | 83.3 | 0.0 |
| | HMMT'25 | **73.3** | 70.0 | -3.3 |
| | *Avg* | *85.8* | *84.2* | *-1.6* |
| | ***Overall*** | *74.6* | *70.2* | *-4.4* |

## B.2. Extended Scaling and Hyperparameter Analysis

### B.2.1. TRACE SCALING ANALYSIS

Figure 5 here in the Appendix extends the scaling analysis from Figure 3(a-c) in the main paper to the complete $4 \times 4$ model-benchmark matrix. Across all 16 configurations, there are 15 out of 16 settings that CDG outperforms majority voting, DeepConf-Mean, and DeepConf-Tail baselines. The advantage persists across different computational budgets (from $L = 8$ to $L = 512$ traces), demonstrating that CDG extracts more discriminative signal per trace regardless of sample size.

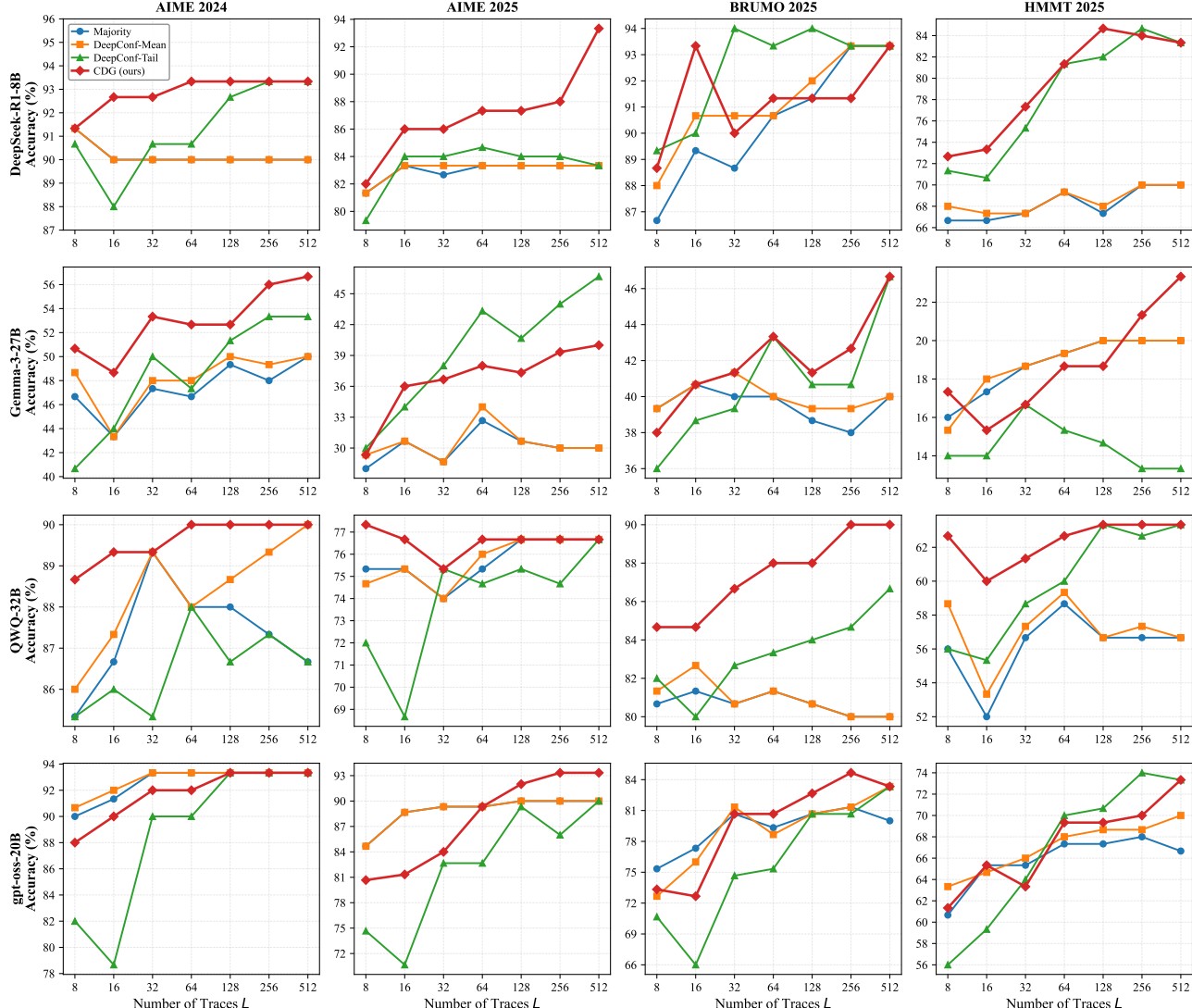

*Figure 5.* Extended scaling analysis: Accuracy vs. number of traces $L$ for all four models (DeepSeek-R1-8B, Gemma-3-27B, gpt-oss-20B, QwQ-32B) across all four benchmarks (AIME 2024, AIME 2025, BRUMO 2025, HMMT 2025). Accuracies are averaged across 5 different random inference runs. CDG consistently outperforms baselines across all configurations.

### B.2.2. SMALL-BUDGET REGIME ($L = 5$–$10$)

As suggested by reviewer, we further include partial results $L = 5$–$10$ for two models to supplement Figure 3. When $L$ is small, answer counts are noisy estimates of answer likelihood, but CDG still offers consistent gains at $L = 5$–$10$, as shown in Table 4.

*Table 4.* Accuracy (%) at small trace budgets $L \in \{5, 10\}$. Best result in each row is in **bold**.

| Model | $L$ | Majority-Vote | DC-Tail | CDG |
|---|---|---|---|---|
| DeepSeek-R1-8B | 5 | 71.5 | 71.3 | **72.2** |
| | 10 | 72.7 | 73.3 | **74.5** |
| Gemma-3-27B | 5 | 28.7 | 29.5 | **31.0** |
| | 10 | 30.7 | 31.0 | **32.8** |

### B.2.3. CDG WEIGHT $\beta$ ABLATION

Figure 6 extends the $\beta$ ablation from Figure 3(d) in the main paper to all four models. The green shaded region indicates the theoretically optimal $\beta$ range, computed as $[0.5r_b, 1.5r_b]$ where $r_b$ is the model-specific ratio derived from our theoretical analysis. The estimated $r_b$ values are: DeepSeek-R1-8B ($r_b = 7.87$), gpt-oss-20B ($r_b = 8.70$), Gemma-3-27B ($r_b = 6.64$), and QwQ-32B ($r_b = 4.39$). This explains why DeepSeek and gpt-oss perform best with larger $\beta$ values ($\beta \approx 10$), while Gemma and QWQ prefer smaller values ($\beta \approx 3\text{-}5$), consistent with our hyperparameter choices in the main experiments.

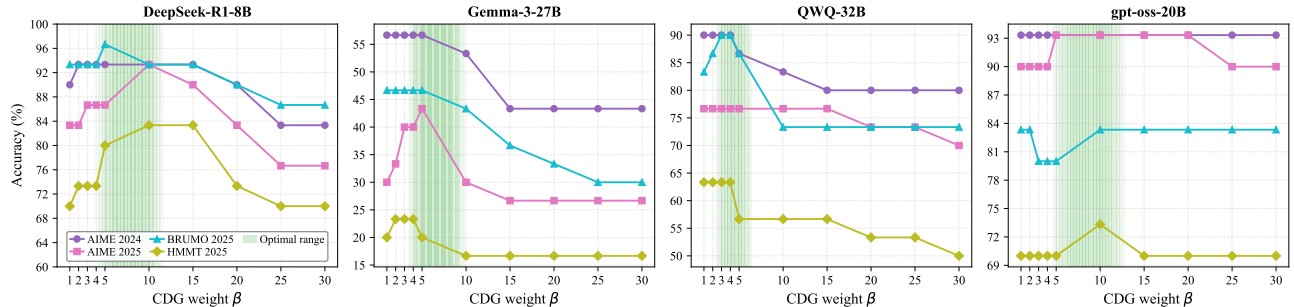

*Figure 6.* Extended $\beta$ ablation: Accuracy vs. CDG weight $\beta$ for all four models across all benchmarks. The green shaded region indicates the theoretically optimal $\beta$ range $[0.5r_b, 1.5r_b]$.

### B.2.4. POSITION PERCENTILE $P$ ABLATION

Figure 7 shows the position percentile ablation study on $P$. Recall that the CDG is computed as $\bar{C}_{\text{tail}} - \bar{C}_{\text{head}}$, where $\bar{C}_{\text{tail}} = \text{mean}(\text{last } P\%)$ and $\bar{C}_{\text{head}} = \text{mean}(\text{first } P\%)$ of the confidence trajectory. This ablation sweeps $P \in \{5, 10, 15, 20, 25, 30\}$ while keeping $\alpha$ and $\beta$ fixed at their optimal values. The relatively flat accuracy curves across all models and benchmarks demonstrate that CDG is robust to the choice of position percentile, requiring no careful tuning of this hyperparameter.

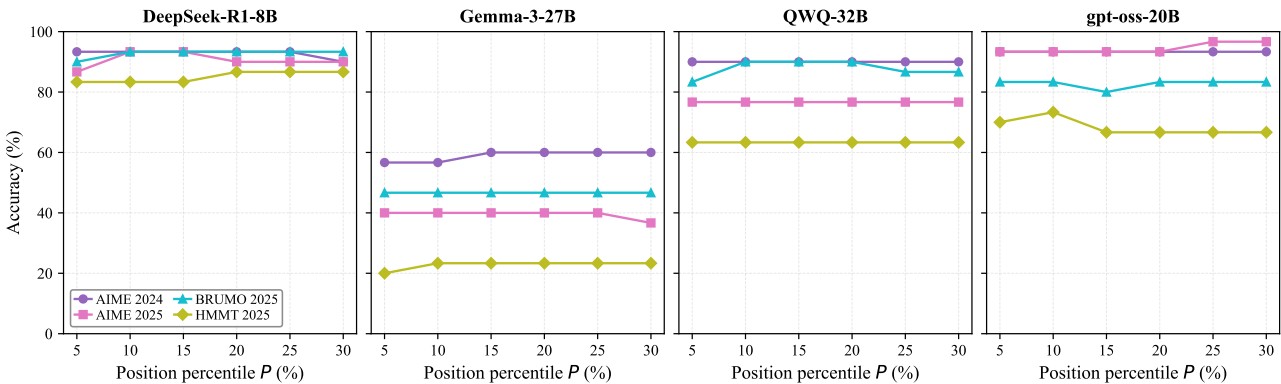

*Figure 7.* Position percentile ablation: Accuracy vs. position percentile $P$ for all four models across all benchmarks. The stable performance across $P \in \{5, 10, 15, 20, 25, 30\}$ demonstrates that CDG is robust to this hyperparameter choice.

### B.3. Statistical Significance of Confidence Dynamics

Table 5 reports the statistical significance of the confidence dynamic gain (CDG) separation between correct and incorrect traces. For each model-dataset pair, we compute the mean CDG for correct traces ($\mu^+$) and incorrect traces ($\mu^-$), then perform a Mann-Whitney U test. All 16 configurations show highly significant differences ($p < 0.001$) with Cohen's $d$ effect sizes ranging from 0.39 to 1.45, confirming that the confidence dynamics pattern is robust and statistically reliable across all models and benchmarks.

*Table 5.* Statistical significance of CDG separation between correct and incorrect traces. $\mu^+$: mean CDG for correct traces; $\mu^-$: mean CDG for incorrect traces; $d$: Cohen's effect size. Significance levels: *** denotes $p < 0.001$ (Mann-Whitney U test).

| Model | Dataset | $N_{\text{corr}}$ | $\mu^+$ | $\mu^-$ | $d$ | Sig. |
|---|---|---|---|---|---|---|
| DeepSeek-R1-8B | AIME 2024 | 13235 | 0.98 | −1.53 | 1.43 | *** |
| | AIME 2025 | 11700 | 0.97 | −1.29 | 1.21 | *** |
| | BRUMO 2025 | 12025 | 0.85 | −1.52 | 1.45 | *** |
| | HMMT 2025 | 9017 | 0.73 | −0.96 | 0.92 | *** |
| Gemma-3-27B | AIME 2024 | 4816 | 0.72 | −1.53 | 0.76 | *** |
| | AIME 2025 | 3783 | 1.39 | −2.34 | 1.12 | *** |
| | BRUMO 2025 | 5502 | 0.33 | −2.37 | 0.86 | *** |
| | HMMT 2025 | 1656 | −0.04 | −3.06 | 1.02 | *** |
| QwQ-32B | AIME 2024 | 12243 | 2.89 | −0.80 | 1.26 | *** |
| | AIME 2025 | 10421 | 2.66 | −0.39 | 1.12 | *** |
| | BRUMO 2025 | 11717 | 2.39 | −1.02 | 1.29 | *** |
| | HMMT 2025 | 7134 | 1.71 | −1.00 | 1.12 | *** |
| gpt-oss-20B | AIME 2024 | 11884 | 3.24 | 2.01 | 0.56 | *** |
| | AIME 2025 | 10847 | 3.13 | 1.74 | 0.54 | *** |
| | BRUMO 2025 | 9891 | 2.97 | 2.12 | 0.39 | *** |
| | HMMT 2025 | 7933 | 2.46 | 1.41 | 0.47 | *** |

### B.4. Statistical Significance of CDG vs Baseline Methods

Table 6 reports the statistical significance of CDG's improvement over baseline methods across the $4 \times 4$ model-dataset grid. We use a paired Wilcoxon signed-rank test (one-sided, testing CDG > baseline) with 112 paired observations (4 models $\times$ 4 datasets $\times$ 7 trace counts). The Win/Tie/Loss (W/T/L) columns report the number of configurations where CDG outperforms, ties, or underperforms each baseline.

CDG significantly outperforms all three baselines: Majority voting ($p < 10^{-11}$), DeepConf-Mean ($p < 10^{-10}$), and DeepConf-Tail ($p < 10^{-10}$). The effect sizes are medium (Cohen's $d \approx 0.7$), indicating practically meaningful improvements. Notably, CDG wins in approximately 70% of all comparisons against DeepConf-Tail (the strongest baseline), with a mean accuracy improvement of $+2.36\%$.

Per-model analysis shows consistent results: all four models exhibit significant improvement over DeepConf-Tail, with QwQ-32B showing the largest effect ($d = 1.54$, 24 wins, 0 losses) and Gemma-3-27B showing the smallest but still significant effect ($d = 0.44$, $p = 0.013$).

### B.5. Question-Level Paired Analysis

To complement the aggregate significance tests in Tables 5 and 6, we report a question-level paired analysis at $L = 512$ that examines CDG's behavior on a per-question basis.

**CDG vs. Majority Voting.** Across all 480 questions (4 models $\times$ 4 datasets $\times$ 30 questions), CDG produces a correct answer on 358 questions (74.6%) compared to 332 (69.2%) for majority voting, an improvement of $+5.4\%$ that is highly significant under a paired $t$-test ($t = 4.70$, $df = 479$, $p < 0.0001$). More importantly, the effect is not merely an aggregate shift: of the 32 questions where the two methods disagree, CDG is correct on 29 while majority voting is correct on only 3. That is, conditional on disagreement, CDG selects the right answer 91% of the time, indicating that the gain is concentrated on the questions for which the choice of voting rule actually matters. Against the stronger baseline DeepConf-Tail, CDG wins 15 of 22 disagreements (68%).

*Table 6.* Statistical significance of CDG improvement over baseline methods. Wilcoxon signed-rank test (one-sided, CDG > baseline). W/T/L: Win/Tie/Loss counts. $d$: Cohen's effect size. Significance: $^{***}p < 0.001$, $^{**}p < 0.01$, $^{*}p < 0.05$.

| Comparison | $N$ | Mean $\Delta$ (%) | W/T/L | $p$-value | $d$ |
|---|---|---|---|---|---|
| *Aggregate (all model-dataset pairs)* | | | | | |
| CDG vs Majority | 112 | +3.12 | 82/14/16 | $2.4 \times 10^{-12}$ *** | 0.77 |
| CDG vs DeepConf-Mean | 112 | +2.71 | 78/15/19 | $7.7 \times 10^{-11}$ *** | 0.68 |
| CDG vs DeepConf-Tail | 112 | +2.36 | 79/17/16 | $6.6 \times 10^{-11}$ *** | 0.73 |
| *Per-model (CDG vs DeepConf-Tail)* | | | | | |
| DeepSeek-R1-8B | 28 | +1.33 | 17/5/6 | $6.7 \times 10^{-3}$ ** | 0.49 |
| Gemma-3-27B | 28 | +1.76 | 19/3/6 | $1.3 \times 10^{-2}$ * | 0.44 |
| QwQ-32B | 28 | +3.10 | 24/4/0 | $8.8 \times 10^{-6}$ *** | 1.54 |
| gpt-oss-20B | 28 | +3.24 | 19/5/4 | $1.2 \times 10^{-4}$ *** | 0.88 |

**Per-question re-test of CDG separation.** Table 5 pools all individual traces when comparing the CDG scores of correct and incorrect traces. To verify that this separation is not driven by a few highly sampled questions, we additionally aggregated at the question level: for each question, we compute the mean CDG score of its correct traces and the mean of its incorrect traces, and then run the significance test *across questions* ($n = 30$ per model–dataset setting). Under this stricter aggregation, all 16 model–dataset configurations remain significant at $p < 0.01$, confirming that the correct-vs-incorrect CDG gap is a per-question phenomenon rather than a sample-size artifact.

## B.6. Trace Length Ablation

Table 7 reports the CDG-vs-majority-voting accuracy improvement (averaged across benchmarks) corresponding to the **Trace Length** analysis in the main paper. For each question, the 512 traces are split into a short pool (256 shortest traces) and a long pool (256 longest traces), and CDG is run independently on each pool. CDG retains a positive margin over majority voting in both pools, indicating that its advantage is driven by the confidence dynamics along the reasoning trajectory rather than by trace-length differences.

*Table 7.* Average accuracy improvement (%) of CDG over majority voting on the short and long trace pools (256 traces each, per question).

| Model | Short (CDG vs Maj) | Long (CDG vs Maj) |
|---|---|---|
| DeepSeek | +4.2% | +4.2% |
| Gemma-3 | +3.3% | +5.0% |

## B.7. GPQA-Diamond Results

To demonstrate that CDG generalizes beyond mathematical reasoning, we evaluate on **GPQA-Diamond** (Rein et al., 2024), a graduate-level multiple-choice benchmark spanning physics, chemistry, and biology, using **DeepSeek-R1-8B** as the underlying reasoning model. Table 8 shows that CDG continues to outperform majority voting and both DeepConf variants on this non-mathematical reasoning benchmark.

We note that our absolute numbers on GPQA-Diamond differ somewhat from those reported in Fu et al. (2025). We conjecture this gap may stem from differences in the prompts used to elicit and extract final answers from the reasoning traces during evaluation; as the corresponding prompts for GPQA-Diamond are not publicly released, we are unable to exactly reproduce that pipeline. Importantly, the relative ordering of methods reported here is obtained under a single, consistent evaluation pipeline applied uniformly to all baselines and to CDG, so the comparison in Table 8 remains internally consistent.

*Table 8.* Accuracy (%) on GPQA-Diamond (Rein et al., 2024) with DeepSeek-R1-8B (DeepSeek-AI, 2025). Best result in **bold**. DC-Mean/DC-Tail denote DeepConf-Mean/Tail with top $10\%$ filtering (Fu et al., 2025); "Majority" is Majority Vote (Self-Consistency) (Wang et al., 2022).

| Majority | DC-Mean | DC-Tail | CDG (ours) |
|---|---|---|---|
| 68.2 | 68.7 | 70.7 | **72.2** |

## C. Implementation Details

### C.1. Sampling Hyperparameters

Table 9 summarizes the sampling hyperparameters used for each model during trace generation. These settings follow the configurations from the respective model documentation and prior work (Fu et al., 2025). We set hyperparameter logprobs = 20 for vLLM across all models.

*Table 9.* Sampling hyperparameters for trace generation used in all experiments.

| Model | Temperature | Top-p | Top-k | Max Tokens |
|---|---|---|---|---|
| DeepSeek-R1-8B | 0.6 | 0.95 | – | 64,000 |
| Gemma-3-27B | 0.6 | 0.95 | 40 | 8,192 |
| QwQ-32B | 0.6 | 0.95 | 20 | 32,768 |
| gpt-oss-20B | 1.0 | 1.0 | 40 | 130,000 |

### C.2. CDG Hyperparameters

For CDG voting, we use the following hyperparameters across all experiments:

- Count dampening exponent: $\alpha = 0.5$

- Position percentile for gradient: $P = 10\%$

- CDG weight $\beta$: $\beta = 10$ for DeepSeek-R1-8B and gpt-oss-20B; $\beta = 3$ for Gemma-3-27B and QwQ-32B

### C.3. Prompts

For DeepSeek-R1-8B, we use the official system prompt in Chinese as specified in the model documentation. For all other models (Gemma-3-27B, QwQ-32B, gpt-oss-20B), we append the following instruction to each question:

```
Please reason step by step, and put your
final answer within \boxed{}.
```

### C.4. Confidence Extraction

Token-level confidence scores are computed following the definition in Eq. (2) of the main paper. We store the top-$K = 20$ logprobs per token position and compute $C_t = -\frac{1}{K} \sum_{j \in \mathcal{K}_t} \log p(y_t = j \mid x, y_{<t})$ for each token $t$, where $\mathcal{K}_t$ denotes the set of top-$K$ tokens at position $t$. The confidence trajectory $\{C_t\}_{t=1}^{T}$ is then partitioned into $N = 10$ position-normalized bins for CDG computation.

### C.5. Answer Extraction and Evaluation

Final answers are extracted from the `\boxed{}` format in model outputs. For mathematical equivalence checking, we use the Dynasor package's `math_equal` function, which handles symbolic equivalence (e.g., $\frac{1}{2} = 0.5$).

## D. Computational Resources

- **Hardware**: $8 \times$ NVIDIA H100 GPUs (80 GB VRAM each)

- **Total traces**: $512 \times 30 \times 4 = 61,440$ traces per model (512 traces $\times$ 30 questions $\times$ 4 benchmarks)

- **Total compute**: Approximately 8,600 GPU hours, including development iterations, baseline recreation and preliminary experiments. A single reproduction run requires significantly less compute.

## E. Dataset Details

We evaluate on four mathematical reasoning benchmarks, each containing 30 problems with verifiable ground-truth answers:

- **AIME 2024** (Balunović et al., 2025): American Invitational Mathematics Examination problems from 2024 (both AIME I and II combined). Source: `MathArena/aime_2024_I`, `MathArena/aime_2024_II`.

- **AIME 2025** (Balunović et al., 2025): American Invitational Mathematics Examination problems from 2025. Source: `MathArena/aime_2025`.

- **HMMT 2025** (HMMT, 2025): Harvard-MIT Mathematics Tournament problems from February 2025. Source: `MathArena/hmmt_feb_2025`.

- **BRUMO 2025** (Brumo, 2025): Bulgarian Mathematical Olympiad problems from 2025. Source: `MathArena/brumo_2025`.

All datasets feature competition-level mathematics problems requiring multi-step reasoning, making them ideal for evaluating confidence dynamics in extended reasoning traces.

