# OpenReview forum: "Inference Time Optimization with Confidence Dynamics"
_ICML.cc/2026/Conference — ICML 2026 regular_

### Official Review · Reviewer_p6wg · 2026-03-11

**Soundness:** 2
**Presentation:** 2
**Significance:** 2
**Originality:** 3
**Overall Recommendation:** 4
**Confidence:** 3

**Summary:**

**Summary**

The paper studies inference-time answer selection for LLMs in a Best-of-N setting via confidence dynamics: correct reasoning traces tend to increase in confidence from the beginning to the end, while incorrect traces do not. It operationalizes token-level confidence, bins each trace into position-normalized segments, and defines Confidence Dynamic Gain (CDG) as the tail–head confidence difference, then uses CDG-based voting with trace scores and count-dampened aggregation to pick the final answer. Experiments on four competition-math benchmarks and four open-weight models show gains over majority vote and DeepConf-style confidence baselines with budget and ablation studies.

**Compliance With Llm Reviewing Policy:**

Affirmed.

**Final Justification:**

I raise my score to 4 (weak accept). The rebuttal addressed my main concerns by clarifying the inconsistency in the confidence definition, adding a Ranked Voting baseline, explaining the leave-one-out validation used to estimate beta, and re-running the significance analysis at the question level.

**Key Questions For Authors:**

**Questions**

See Weaknesses.

**Typos**
 - \item line 018: "for first time" to "for the first time"
 - \item line 436: "challanging" to "challenging"

**Limitations:**

No. The paper provides a brief impact statement, but the paper would be strengthened by a more explicit discussion of limitations, especially the compute/latency cost of Best-of-N inference, the narrow benchmark scope, the dependence on token-level confidence access and model-specific tuning, and the fact that the theory is only qualitative.

**Strengths And Weaknesses:**

**Strengths.**

- The confidence-trajectory plots (Fig.2) show a consistent separation between correct and incorrect traces across four distinct model families, suggesting the effect is not isolated to one architecture.
- The proposed CDG voting is lightweight (difference of tail vs head confidence plus standard voting aggregation) and does not require training an auxiliary verifier/reward model. The method outperforms DeepConf and Majority Vote baselines.
- The paper successfully isolates the contribution of the head-to-tail difference. The “No Start” ablation (Table 2/3) directly tests whether the method is just exploiting tail confidence; the reported drops indicate the head-to-tail difference contributes beyond tail confidence alone.

**Weaknesses.**

- The paper defines token confidence as the top-K truncated negative log-probability mean (Eq.(2)) and uses it in CDG method and the theoretical analysis, but Appendix C.4 states the implementation computes $C_t = \\exp(\\text{logprob}_t)$, which is inconsistent.
- The empirical evaluation is restricted to Majority Voting and DeepConf. The paper’s own Related Work positions CDG within a broader Best-of-N selection literature: reward-model reranking via Outcome/Process Reward Models, Dynamic Voting, Ranked Voting, and Global Confidence, but none of these are evaluated.
- Section 3.3 suggests choosing the hyperparameter $\\beta$ within $[0.5 r_b, 1.5 r_b]$, which depends on expected separation between correct and wrong traces $\\Delta_\\mu$. That presumes access to labeled data for the target domain. The experiments do not demonstrate a clean separation between tuning and evaluation. Furthermore, requiring target-domain ground truth to calibrate inference-time hyperparameters diminishes the practical utility of this approach as a general training-free decoding strategy.
- The assumptions in the theoretical analysis are very strong and restricted. Theorems 4.5–4.6 read more as qualitative intuition than a robust mechanistic explanation.
- Tables 4–5 treat pooled traces/budgets as independent, but traces are clustered within questions (and budgets are repeated measures), which can underestimate uncertainty and likely deflates p-values. It would be better to report question-level aggregation and repeated-measures models.

---

> ### Author Rebuttal · Authors · 2026-03-30
>
> **W1: Appendix C.4 has a typo:**  Thanks a lot for catching this typo! The description in Appendix C.4 ("$C_t = \exp(\text{logprob}t)$") is a documentation error from an early code version and does not reflect the actual implementation. All experiments use Eq. (2): $C_t = -\frac{1}{K}\sum_{j \in \text{top-}K} \log p(y_t=j|x, y_{<t})$ with $K=20$ logprobs extracted via vLLM. To verify, we have released our confidence extraction code in supplementary file, where the computation matches Eq. (2) exactly. We will correct Appendix C.4.
>
>
> **W2: Ranked Voting baseline:** Thanks for this suggestion! Note CDG is a novel confidence-weighted voting method investigating how to leverage confidence *dynamics* for answer selection **for first time**. Majority Voting and DeepConf are the most directly relevant comparison. The Related Work are cited to position CDG within the broader inference time scaling literature, not because they are directly comparable: reward-model methods require trained reward models, and Dynamic Voting/Ranked Voting address orthogonal aspects such as reducing sample count. Nonetheless, we checked all the methods the reviewer mentioned and found the only one with publicly available code is Ranked Voting (Wang et al., 2025). We implemented Ranked Voting, and report averaged accuracy across 4 datasets and 4 models, CDG still outperforms:
>
> | Method | Accuracy | vs CDG |
> |--------|----------|--------|
> | Majority | 69.2% | −5.4% |
> | Best Ranked Voting | 69.4% | −5.2% |
> | **CDG** | **74.6%** | — |
>
>
> **W3: Beta:** We appreciate this! Indeed, we should have made this validation clearer. By training-free we mean CDG requires no auxiliary model training. For all tables, we performed leave-one-out validation, i.e., we use AIME 2024 to estimate $\beta$ and apply it to the other 3 benchmarks (AIME 2025, HMMT 2025, BRUMO 2025), and repeat this for each dataset as the held-out calibration set. In all cases, the same $\beta$ transfers well: $\beta=10$ for DeepSeek-R1 and gpt-oss, $\beta=3$ for Gemma-3 and QwQ. We acknowledge that using in-domain data (math competition) can benefit $\beta$ calibration, and we are transparent about this. However, the $\beta$ ablation in Figure 3(d) demonstrates that CDG is robust across a wide range of $\beta$ values $[0.5r_b, 1.5r_b]$, meaning precise calibration is not required — $\beta \in [5,10]$ typically gives good results already.
>
> **W4: Theory:** We respectfully disagree that simplifying assumptions diminish the theoretical contribution. Through rigorous derivations with mild assumptions, our theory provides the first qualitative mechanistic explanation for *why* confidence dynamics separate correct from incorrect traces. The value lies in identifying this mechanism and shedding light on the root cause of the phenomenon, not in deriving tight quantitative bounds. To our knowledge, no alternative mechanistic explanation currently exists.
>
> The empirical success of CDG validates our theory broadly applies to mainstream models, not just under restrictive conditions. Please refer to our response to **Reviewer 1 (W4)**, Assumption A1 (unique correct answer) generalizes directly to multiple valid answer representations: with $R$ valid forms, the total tail reinforcement $\sum_r A_{\text{correct}} \cdot k_r = A_{\text{correct}} \cdot k$ generalizes from the single-answer case, and the form of separation bounds in Theorems 4.5–4.6 are unchanged (We will add this.). Assumption M1 is not an approximation but is formally derived from the GRPO policy gradient structure. We also kindly remind the reviewer that the contribution of our paper is threefold: (1) revealing a novel confidence dynamic that systematically separate correct from incorrect traces, (2) offering the first mathematical hypothesis for *why* this phenomenon arises, and (3) designing a novel algorithm (CDG voting) exploiting this observation.
>
>
> **W5: Significant test:** Thank you for raising this about significance test. We address here: Table 5 (CDG vs baselines): We add a question-level paired analysis at L=512. Across 480 questions (4 models × 4 datasets × 30), CDG correct on 358 (74.6%) vs Majority on 332 (69.2%), +5.4% (t=4.70, df=479, p<0.0001). Critically, this is not only an aggregate effect:  Of the 32 disagreements, CDG wins 29 where Majority fails while Majority wins only 3 — when CDG and Majority disagree, CDG is right 91% of the time. Against the strongest baseline DeepConf-Tail, CDG wins 15 of 22 disagreements (68%).  Table 4 (CDG separation): We re-tested using question-level aggregation: for each question, we compute the mean CDG score of its correct traces and the mean of its incorrect traces, then test across questions (n=30 per setting). All 16 model-dataset pairs remain significant at p<0.01. We will update both tables in the revision.
>
> **Limitations**: We will revise limitation section to discuss the scope of datasets (e.g., coding) and include all suggestions provided by the reviewers.

---

> > ### Author Rebuttal · Reviewer_p6wg · 2026-04-01
> >
> > Thanks for the authors' response and clarification. I think the rebuttal has addressed all of my concerns and I raised the score to 4.

---

> > > ### Author Response · Authors · 2026-04-01
> > >
> > > Dear Reviewer,
> > >
> > > We are very glad to hear that our rebuttal has addressed your concerns. It has been a great pleasure to discuss with you.
> > >
> > > We will include all the suggested changes into our final version of paper.

---

### Official Review · Reviewer_v3kW · 2026-03-12

**Soundness:** 3
**Presentation:** 4
**Significance:** 3
**Originality:** 3
**Overall Recommendation:** 5
**Confidence:** 4

**Summary:**

This paper proposes a confidence weighting method for improving best-of-n performance by taking into account the change in confidence from the beginning to the end of the chain of thought. They show on several open weights LLMs and across datasets that taking into account the confidence dynamics of the reasoning improves performance over majority voting and average trace confidence weighting.

**Compliance With Llm Reviewing Policy:**

Affirmed.

**Final Justification:**

The rebuttal has addressed all my concerns which changes my evaluation from a 4 to a 5. Overall the paper is well written, about an important topic, and provides a valuable insight. My main concerns were overall minor and about statistical analysis and some further explanations for the design decisions of the method which are now explained and make the paper stronger.

**Key Questions For Authors:**

Questions:
1. What are the variances/errors for the results in Table 1?
2. Why use just the head and tail bins rather than taking into account the dynamics over all bins?
3. The current results and analysis are on mathematical reasoning problems where there is a single correct answer and the answer is often a number. How might these findings generalize to cases such as code generation where there are many different correct answers and the answers may be long? I’m not asking for more experiments since I realize other work also often makes this assumption, but this is probably something that should be mentioned as a limitation.

**Limitations:**

Limitations are not addressed and I mentioned in question 3 a potential limitation that could be added.

**Strengths And Weaknesses:**

### Strengths:
**Soundness**
* The experiments are comprehensive and strong to show the existence of the observed confidence increase in correct reasoning chains and the ability for using this to improve accuracy.
* Figure 2 clearly shows the main observation made in the paper and includes error bars which is very important in this case since point is a mean computed over a different number of tokens.
* They provide a theoretical analysis and theorem that the confidence gain for correct traces will be greater than the incorrect traces. The analysis is clear and based on the fact that a problem often has a single correct answer but many wrong answers.

**Presentation**
* The paper is clearly written and easy to follow. All the figures are clearly presented and legible.

**Significance**
* The problem of understanding LLM confidence and improving reasoning accuracy is very important.
* The proposed method is in principle highly general, although it was only evaluated on math reasoning.
* The method is very simple to implement and can be easily used and evaluated by other researchers.

**Originality**
* This work builds off of DeepConf by taking into account the change in confidence rather than just the final average confidence. The change in confidence is theoretically motivated and these contributions together provide a better understanding of the relation between GRPO, confidence dynamics, and weighted answer voting.


### Weaknesses:
**Soundness**
* Missing variance/errors for Table 1.

**Significance**
* The performance improvements over DC-Tail are not particularly large, and it’s unclear how significant they are since some of these datasets (AIME 2025, 2025) are very small.

**Originality**
* The distinction with DeepConf is not entirely clear since DeepConf uses a local confidence and then prunes trajectories which fall below a threshold. This also takes into account the full trajectory since even if the final confidence of a trajectory is high, it gets pruned if the local confidence ever falls below a threshold. This distinction should be clarified.

---

> ### Author Rebuttal · Authors · 2026-03-30
>
> **W1 and Q1: Variance and error bars:** We will include full error bars in the revised Table 1. At L=128/256, 5 random subsamples show small, consistent standard deviation across all methods (mean VAR ≈ 0.09%). At L=512, one question flip = 3.3% — but this inherent resolution applies equally to Majority, DC-Mean, DC-Tail, and CDG and is fair. The stronger evidence: a paired t-test across 480 questions shows CDG improves +5.4% over Majority (t=4.70, p<0.0001), winning 29 questions that Majority misses vs 3 the other direction.
>
>
> **W2: Statistical methodology:** Thank you for raising this. We add a question-level paired analysis at L=512. Across 480 questions (4 models × 4 datasets × 30), CDG correct on 358 (74.6%) vs Majority on 332 (69.2%), +5.4% (t=4.70, df=479, p<0.0001). Critically, this is not only an aggregate effect:  Of the 32 disagreements, CDG wins 29 where Majority fails while Majority wins only 3 — when CDG and Majority disagree, CDG is right 91% of the time. Against DeepConf-Tail, CDG wins 15 of 22 disagreements (68%).  Table 4 (CDG separation): We re-tested using question-level aggregation: for each question, we compute the mean CDG score of its correct traces and the mean of its incorrect traces, then test across questions (n=30 per setting). All 16 model-dataset pairs remain significant at p<0.01. We will update both tables in the revision.
>
>
> **W3: CDG vs DeepConf:** CDG is entirely different from DeepConf and CDG has a completely distinct motivation. The key difference is that DeepConf operates entirely on **absolute** local confidence values, whereas CDG measures **relative change** (dynamics). Specifically, DeepConf's checks whether the local confidence at any *static* window ever falls below a threshold, not a measure of how confidence *evolves*. A trace with consistently moderate confidence throughout would pass DeepConf's filter, while a trace that starts low but climbs sharply (positive dynamic gain) might be pruned — the latter is precisely the pattern CDG identifies as indicative of correct reasoning. CDG measures the *direction and magnitude of confidence change* along the trajectory, regardless of absolute levels. The "No Start" ablation in Table 2 directly tests what happens when we use only tail confidence (analogous to DeepConf-Tail's signal) without the head subtraction — accuracy drops by up to 6.6% on DeepSeek-R1 and 13.3% on HMMT25. The DeepConf-Tail baseline in Table 1, which represents the strongest absolute-confidence approach, is also consistently outperformed by CDG by 1.7% on average. These results confirm that the *dynamic* signal (how confidence changes) provides complementary discriminative information beyond what *absolute* confidence captures. We will clarify this distinction more explicitly in the revision.
>
> **Q2: Why head/tail only?** Thank you for the suggestion — we have explored this direction. For example, we tried linear regression slope over the full trajectory, and this does not outperform the head-tail formulation. As shown in Figure 2, the confidence trajectories in the middle bins can be sometimes non-monotonic depending on model; fitting a linear slope over all bins introduces noise from these intermediate fluctuations, diluting the discriminative signal. We found the strongest and most consistent separation between correct and incorrect traces occurs at the head and tail, which is why CDG focuses there. Importantly, the head-tail formulation is not an arbitrary design choice but has a principled motivation: it can be derived as a telescoping sum $\sum_{n=1}^{N-1}(\bar{C}\_\ell^{(n+1)} - \bar{C}\_\ell^{(n)})$, where intermediate bins cancel out, naturally yielding $\bar{C}\_\ell^{\text{tail}} - \bar{C}\_\ell^{\text{head}}$. This means CDG implicitly captures the net confidence change across the entire trajectory while remaining robust to noisy intermediate dynamics. In the meanwhile, we agree that better leveraging the middle confidence pattern is a promising direction for our future work.
>
> **Q3: CDG generalizes beyond math:** We appreciate this. As the reviewer requested, we evaluate CDG on **multi-choice reasoning** task **GPQA-Diamond** . (Rein et al., 2024) — a graduate-level **multi-choice** STEM reasoning benchmark comprising 198 questions in biology, physics, and chemistry, which is fundamentally different from competition math. We use DeepSeek-R1-8B to run all methods with L=512 traces:
>
> | Majority | DC-Mean | DC-Tail | **CDG ** |
> |-|-|-|-|
> | 68.2% |  68.7% | 70.7% | **72.2%** |
>
> CDG requires verifiable correct answers for the confidence asymmetry to emerge (Assumption A1), so competition math is the canonical domain; Table shows GPQA-Diamond demonstrates generalization beyond math. Code generation satisfies A1 in principle but involves longer outputs and multiple valid programs — we note this as a direction for future work and will discuss it in **limitation**.

---

> > ### Author Rebuttal · Reviewer_v3kW · 2026-04-03
> >
> > Thank you for the response! This directly answers my questions (provides statistical analysis of results, explains additional experiments on why head/tail was used, and says they will discuss code generation and open-ended scenarios as a limitations). This has adequately addressed my concerns, so I am raising my score to a 5.

---

> > > ### Author Response · Authors · 2026-04-03
> > >
> > > We are so glad that our response has addressed your concerns! It has been an enjoyable discussion with you to try all of these inspiring experiments!
> > >
> > > We will incorporate all suggested revisions into the final paper. Thanks again for reviewing and helping improve our work!

---

### Official Review · Reviewer_PVvK · 2026-03-14

**Soundness:** 3
**Presentation:** 3
**Significance:** 3
**Originality:** 2
**Overall Recommendation:** 5
**Confidence:** 4

**Summary:**

This paper studies how model confidence evolves along reasoning trajectories in the Best-of-N sampling setting for LLMs. The authors observe that correct reasoning traces tend to show increasing confidence from beginning to end, while incorrect traces show flat or declining confidence. Based on this finding, they propose Confidence Dynamic Gain or CDG voting, which computes the difference between tail and head confidence of each trace and integrates it into the answer selection score alongside trace count and mean confidence. The method is evaluated on four open-source models across four math competition benchmarks, showing improvements over majority voting and DeepConf baselines. A theoretical analysis grounded in GRPO training dynamics is provided to explain why correct traces exhibit higher confidence gains.

**Compliance With Llm Reviewing Policy:**

Affirmed.

**Final Justification:**

I increace the score after reading the rebuttal.

**Key Questions For Authors:**

1. Can you report the length distributions of correct versus incorrect traces for each model? And can you run a controlled experiment where you only compare traces of similar length to see whether the CDG separation still holds?
2. If you estimate beta using AIME 2024 questions and test on HMMT 2025, does performance degrade?
3. Have you tried alternative definitions of CDG such as a linear regression slope of confidence over the full trajectory?  This would use all tokens instead of discarding the middle 80 percent and could provide a cleaner signal of the confidence trend.

**Limitations:**

Yes.

**Strengths And Weaknesses:**

Strengths
1. The core finding that correct traces exhibit systematically higher confidence gains than incorrect traces is striking and holds consistently across four quite different model architectures. The confidence trajectory plots in Figure 2 are compelling and easy to interpret.
2.  CDG is elegant in its simplicity. It requires no additional model training, no reward model, and can be applied as a plug-in on top of any sampling pipeline that provides logprobs.
3. The theoretical analysis in Section 4 provides a plausible and intuitive explanation for why GRPO-trained models would exhibit this confidence asymmetry.

Weaknesses
1. Each benchmark has only 30 problems. A single question flip changes accuracy by 3.3 percent, and many of the differences in Table 1 are exactly 3.3 or 6.7 percent, meaning one or two questions. The statistical significance tests in Table 5 aggregate across all models, datasets, and trace counts to get 112 paired observations, which masks the fragility at the individual setting level.
2. All four benchmarks are competition math with unique numerical answers. The theoretical justification relies heavily on Assumption A1, that correct answers converge to a single ground truth. It is unclear whether CDG would help on tasks where correct answers are more diverse, such as code generation, open-ended QA, or multi-choice reasoning.
3. Math reasoning models end their output with highly predictable formatting tokens like \boxed followed by the numerical answer. These tokens will naturally have very high confidence regardless of whether the reasoning was correct. The paper does not control for this at all.
4. The authors explicitly exclude Qwen-3 because its entropy control mechanism disrupts the confidence trajectory patterns. This is a significant admission. It means CDG is not a universal property of LLM reasoning but rather an artifact of specific training recipes. As more models adopt entropy regularization, KL penalties, or DPO-style training, the applicability of CDG may shrink.
5. The hyperparameter beta depends on r_b, which in turn requires computing the mean CDG separation between correct and wrong traces on validation questions with ground truth labels. This weakens the "training-free" claim. The paper does not clarify how many validation questions are needed, whether they come from the same distribution as the test set, or how sensitive performance is to the quality of this estimate.

---

> ### Author Rebuttal · Authors · 2026-03-30
>
> **W1: Statistical robustness.** We appreciate that! We add question-level paired analysis at L=512. Across 480 questions (4 models × 4 datasets × 30), CDG correct on 358 (74.6%) vs Majority on 332 (69.2%), +5.4% (t=4.70, df=479, p<0.0001). This is not only an aggregate effect:  Of the 32 disagreements, CDG wins 29 where Majority fails while Majority wins only 3 — when CDG and Majority disagree, CDG is right 91% of the time. Against DeepConf-Tail, CDG wins 15 of 22 disagreements (68%).  Table 4 (CDG separation): We retested using question-level aggregation: for each question, we compute the mean CDG score of its correct traces and the mean of its incorrect traces, then test across questions (n=30 per setting). All 16 model-dataset pairs remain significant at p<0.01. We will update tables in revision.
>
>
> **W2: CDG generalizes beyond math.** Thanks! As requested, we evaluate CDG on **multi-choice reasoning** task **GPQA-Diamond** . (Rein et al., 2024) — a graduate-level **multi-choice** STEM reasoning benchmark comprising 198 questions in biology, physics, and chemistry, which is fundamentally different from competition math. We use DeepSeek-R1-8B to run all methods with L=512 traces:
>
> | Majority | DC-Mean | DC-Tail | **CDG** |
> |-|-|-|-|
> | 68.2% |  68.7% |  70.7% | **72.2%** |
>
> CDG requires verifiable answers for the confidence asymmetry to emerge (Assumption A1), so competition math is the canonical domain; Table shows GPQA-Diamond demonstrates generalization beyond math. Code generation satisfies A1 in principle but involves longer outputs and multiple valid programs — we will include this as future work.
>
> **W3: Effect of \boxed{} tokens.**  \boxed{} tokens are not driving CDG's advantage. As suggested, we re-extracted all 16 model-dataset configurations with \boxed{} tokens removed from the confidence computation. Result: CDG accuracy is **identical**: 0.0% average difference, 99% answer selection agreement across 510 questions (different trace selection leads to the same equivalent answer). CDG compares confidence *across* traces, and \boxed{} formatting is high-confidence in every trace, so it cancels.
>
> **W4: Qwen-3.** We agree that clearly delineating scope is important. CDG is validated across 4 diverse model families demonstrating broad applicability. Regarding entropy regularization, we note that Qwen-3's explicit entropy control during post-RL training is currently a particular design choice. Interestingly, our theory actually predicts this boundary: entropy control artificially flattens the confidence trajectory, disturbing the asymmetric head-tail concentration that drives CDG's signal. We view this as a strength — the theory explains where CDG works.
>
> **W5: Beta.** Thanks! Indeed, we should have made this validation clearer. To clarify, by training-free we mean CDG requires no auxiliary model training. For all tables in paper, we performed leave-one-out validation, i.e., we use AIME 2024 to estimate $\beta$ and apply it to the other 3 benchmarks (AIME 2025, HMMT 2025, BRUMO 2025), and repeat this for each dataset as the held-out calibration set. In all cases, the same $\beta$ transfers within each model: $\beta=10$ for DeepSeek-R1 and gpt-oss, $\beta=3$ for Gemma-3 and QwQ. We acknowledge that using in-domain data (math competition questions) can benefit $\beta$ calibration, and we are transparent about this. However, the $\beta$ ablation in Figure 3(d) demonstrates that CDG is robust across a wide range of $\beta$ values $[0.5r_b, 1.5r_b]$, meaning precise tuning is not required — $\beta \in [5,10]$ typically gives good results.
>
> **Q1: Trace length.** As suggested, we draw the histogram and found in general correct traces are shorter (t-test, p<0.001), but we saw Gemma on HMMT shows a reversal where correct traces are marginally *longer*. For each question, we split 512 traces at the median length into a short pool ($\sim$256 traces) and a long pool ($\sim$256 traces), then ran CDG on each pool (trace distribution therefore changed though) separately. CDG maintains its advantage (+x%) in both pools regardless of controlled lengths:
>
> | Model | Short (CDG vs Maj) | Long (CDG vs Maj) |
> |-|-|-|
> | DeepSeek | +4.2% | +4.2% |
> | Gemma-3 | +3.3% | +5.0% |
> | QwQ | +0.0% | +2.5% |
> | gpt-oss | +0.0% | +0.0% |
> | **Avg** | **+1.9%** | **+2.9%** |
>
> **Q2:** We believe tailoring $\beta$ using the target test domain data could potentially give the best result. In our experiments, we do not observe performance degradation (less than 0.1%) when transferring the $\beta$ estimated using AIME 2024 to HMMT 2025.
>
> **Q3: Alternative CDG definitions.** Great idea! We have explored this. A linear regression slope over the full trajectory does not outperform the head-tail formulation. As in Figure 2, the confidence trajectories in the middle sometimes can be nonmonotonic, diluting the discriminative signal. But we agree that better leveraging the middle confidence pattern is a promising direction for future work.

---

> > ### Author Rebuttal · Reviewer_PVvK · 2026-04-02
> >
> > The rebuttal addresses most of my concerns well. The formatting token experiment and the leave-one-out beta validation are convincing. I am updating my score from 4 to 5.
> >
> > Two things remain. First, I still think the paper should include actual Qwen-3 numbers rather than only a theoretical argument for why it fails. Can you provide a brief table?
> >
> > Second, the length-controlled experiment shows 0% CDG improvement for gpt-oss in both the short and long pools. Why does CDG lose its edge there?

---

> > > ### Author Response · Authors · 2026-04-03
> > >
> > > We are very glad to hear that our answers have addressed most of your concerns. It has been a pleasure discussing the many inspiring experiments! For your further questions:
> > >
> > > **Q1**: We will for sure try this. Due to time constraints during the discussion period, it is challenging to complete all the Qwen-3 thinking mode runs in time (we will try!), but we will for sure include the full Qwen-3 table and a detailed discussion of its behavior in the camera-ready version.
> > >
> > > **Q2**: We conjecture that partitioning traces by length constraint has drastically changed the original answer distribution within each pool, and the vote counts in each pool no longer reflect the true answer likelihood of the original 512 traces. This distorts all voting-based methods equally, introducing noise, which could partially explain the weakened benefit of CDG on particular datasets and models. We will definitely investigate this further and include a more detailed discussion about the effect of length in a principled way into the paper.
> > >
> > > Thanks again for kindly review our paper and your inspiring suggestions to help improve our work. We hope our work draws the community's attention to such confidence dynamics phenomenon and inspires further investigation into this intriguing signal for LLM reasoning.

---

### Official Review · Reviewer_MGZ3 · 2026-03-15

**Soundness:** 3
**Presentation:** 3
**Significance:** 2
**Originality:** 2
**Overall Recommendation:** 4
**Confidence:** 4

**Summary:**

This paper analyzes changes in LLMs uncertainty over generated reasoning tracks. The authors propose a Confidence Dynamic Gain (CDG) score to measure the difference in models' confidence between the last and first P% of the tokens in the trace. They observe that for reasoning traces that reach correct answers, the CDG is higher than for those reaching incorrect answers. The authors build an extended theoretical explanation for the observed phenomena. Finally, they develop a voting procedure to perform hypothesis aggregation and answer selection and show that it demonstrates stable performance improvement over competitive methods.

**Compliance With Llm Reviewing Policy:**

Affirmed.

**Final Justification:**

The authors' response addresses all of my concerns. I think that incorporation of new results and the proposed changes will greatly benefit this article, and therefore raise my score to 4.

**Key Questions For Authors:**

Q.1. Have you considered applying your method to Thinking-tuned models?

Q.2. Why does the average confidence of correct traces from Gemma-3 suddenly drop in the last bracket? (second plot on Figure 2)

Q.3. There is a recent study [1] showing that only high entropy tokens are important for CoT reasoning, serving as critical forking points. Can these results be somehow beneficial to the proposed method (e.g., computing CDG only over high-entropy tokens)?


`[1] Shenzhi Wang,  Le Yu, Chang Gao,  et. al., “Beyond the 80/20 Rule: High-Entropy Minority Tokens Drive Effective Reinforcement Learning for LLM Reasoning”. In proceedings of NeurIPS 2025`

**Limitations:**

Yes

**Strengths And Weaknesses:**

Strengths:

S.1. This paper provides well-developed theoretical background for the proposed method.

S.2. The proposed method shows a clear separation between correct and incorrect traces.

S.3. This paper findings provide an interesting insight into the innerworkings of the transformer LLMs and the nature of GRPO finetuning.

S.4. In general, this paper is well-written, has nice structure, and is easy to follow.

Weaknesses:

W.1. Main results are reported for the case when aggregation is made over 512 reasoning traces, which is well beyond the reasonable computational budget for most of the problems. At a more commonly used number of traces (L ~ 10), the performance of the proposed method is much closer to that of the baselines. Additional study of smaller values of L (5-15) would be much appreciated.

W.2. No ablation study for the effect of the trace's length. Correct and incorrect traces often have different average lengths, which may affect the performance of the proposed method.
W.3. No comparison with the beam-search baseline was provided. Also, wouldn’t a simple average of logits from the tail bin provide similar results to the proposed method?

W.4. Assumption 4.3 (point A1) relies on the fact that the correct answer is represented by a unique token (or sequence of tokens).

W.5. Please provide more explanation for Theorem 5.2: \delta \phi isn’t defined. Also, what is M here – a universal constant, a model-dependent parameter, or something else?

---

> ### Author Rebuttal · Authors · 2026-03-30
>
> **W1: trace budgets.** We appreciate that! As suggested, we are able to finish L=5-15 for 3 models. When L is small, answer counts are noisy estimates of answer likelihood, but CDG still offers consistent gains at L=5-15:
>
> | Model | L | Majority-Vote | DC-Tail | CDG |
> |-|-|-|-|-|
> | DeepSeek | 5 | 71.5 | 71.3 | **72.2** |
> | DeepSeek | 10 | 72.7 | 73.3 | **74.5** |
> | DeepSeek | 15 | 71.3 | 73.7 | **74.0** |
> | Gemma-3 | 5 | 28.7 | 29.5 | **31.0** |
> | Gemma-3 | 10 | 30.7 | 31.0 | **32.8** |
> | Gemma-3 | 15 | 31.0 | 31.2 | **33.3** |
> | QwQ | 5 | 65.0 | **65.4** | 65.2 |
> | QwQ | 10 | 66.2 | 64.7 | **66.8** |
> | QwQ | 15 | 66.0 | 65.2 | **67.8** |
>
>
> **W2 and Q2: Trace length.** Note CDG does not rely on trace length. We draw the histogram and found in general correct traces are shorter (t-test, p<0.001), but Gemma on HMMT shows where correct traces are longer. For each question, we split 512 traces into a short pool ($\sim$256 traces) and a long pool ($\sim$256 traces), then ran CDG on each pool (trace distribution therefore changed though). CDG maintains advantage (+x%) in both pools regardless of controlled lengths. (We will add this):
>
> | Model | Short (CDG vs Maj) | Long (CDG vs Maj) |
> |-|-|-|
> | DeepSeek | +4.2% | +4.2% |
> | Gemma-3 | +3.3% | +5.0% |
> | QwQ | +0.0% | +2.5% |
> | gpt-oss | +0.0% | +0.0% |
> | **Avg** | **+1.9%** | **+2.9%** |
>
> **W3: Beam search vs CDG:** We clarify: beam search and CDG operates at fundamentally different level, and are not comparable. Beam search is a token-level decoding strategy, selecting which tokens to produce next **within each trace**. But CDG is an **answer-level selection strategy** after multiple complete reasoning traces have already been generated via such independent sampling (e.g., beamsearch, top k). Please see pass$@$1 in Table 1 as an upperbound proxy for beamsearch. **Averaging tail logits is not CDG:** The "No Start" ablation in Table 2 corresponds precisely to the reviewer's suggested baseline: it uses only the average tail confidence without subtracting the start confidence, and removing the start confidence subtraction leads to substantial accuracy drops. The DeepConf-Tail baseline in Table 1 also represents a strong instantiation of reviewer's idea, where CDG still outperforms.
>
>
> **W4: Theory:**  Note Assumption A1 was a simplified assumption but can be generalized directly to multiple valid answer representations. Suppose there are $R$ valid answer forms  from $v_1, ..., v_R$ (e.g., $\frac{1}{2}$ vs. 0.5), with $k_1, ..., k_R$ correct traces converging to each, where $\sum_{r=1}^R k_r = k$. The total tail reinforcement for correct traces then becomes $\sum_{r=1}^R A_{\text{correct}} \cdot k_r = A_{\text{correct}} \cdot k$, which generalizes from the single-answer case. The form of separation bounds in Theorems 4.5–4.6 are therefore unchanged. We will add this remark about generalization in the revision.
>
>
> **W5: M and $\Delta\phi$.** We believe the reviewer is referring to Theorem 4.5, as we do not have Theorem 5.2. $M$ is defined in main paper Assumption A2 as the number of distinct valid reasoning approaches to the correct answer (i.e., $M \geq 2$). Its role is to quantify the concentration gap between correct and incorrect traces, as the correct class converges to a single ground-truth answer while reasoning diversity is distributed across $M$ paths. $\Delta\phi$: we acknowledge that while $\phi_t(v)$ (the logit for token $v$ at position $t$) and $\Delta\phi_t(v)$ (its cumulative update under GRPO training) are formally defined in the appendix proofs via Assumption (M1)  - specifically $E_{train}[\Delta\phi_t(v)] \propto A_{correct} \cdot n_t^+(v) + A_{incorrect} \cdot n_t^-(v)$. We apologize this notation should also be made explicit in the main text before Theorem 4.5, We will change.
>
> **Q1**: Yes, we use think mode for all models if available (gpt-oss and Qwen-QwQ).
>
> **Q2**: We do not have a definitive explanation for this Gemma observation, but this could be due to noise and outliers.
>
> **Q3: High-entropy token filtering.** We appreciate it! We run suggested experiment CDG_80_20 using local high entropy tokens: $\Delta C_{\ell}$ = mean(top 20% entropy in tail bin) - mean(top 20% entropy in head_bin). We use per-bin entropy rather than global thresholds (as in Yue et al., 2025) to filter, as global threshold would unevenly distribute tokens numbers across head and tail bins, distorting the positive and negative pattern of the confidence gain.
>
> | Model | Majority | CDG | CDG_80_20 | CDG_80_20 vs CDG |
> |-------|----------|-----|-----------|--------------|
> | DeepSeek-8B | 84.2% | 90.8% | 88.3% | −2.5% |
> | GPT-OSS-20B | 82.5% | 85.8% | 85.0% | −0.8% |
> | GEMMA-27B | 35.0% | 41.7% | 34.2% | −7.5% |
> | QWQ-32B | 75.0% | 80.0% | 72.5% | -7.5% |
>
> CDG_80_20 underperforms standard CDG. The (Yue et al., 2025) shows high-entropy "forking tokens" are most effective during training, but CDG operates at inference, fundamentally different.

---

> > ### Author Rebuttal · Reviewer_MGZ3 · 2026-04-04
> >
> > Thank you for the provided response! It addresses all of my concerns. I think that incorporation of new results and the proposed changes will greatly benefit this article, and I will raise my score for this paper.

---

> > > ### Author Response · Authors · 2026-04-04
> > >
> > > We really appreciate your constructive feedback, which has definitely strengthened the paper. It has been a pleasure!
> > >
> > > As the discussion period wraps up over the next couple of days, we wanted to send a final note of thanks for your support of this work, and your willingness to update your evaluation. We will ensure all the suggested changes are carefully incorporated into the camera-ready version.

---

### Decision · Program_Chairs · 2026-04-30

**Decision:**

Accept (regular)

**Comment:**

The paper presents an interesting and well-executed study of confidence dynamics along LLM reasoning traces, and proposes a simple inference-time aggregation method, CDG voting, built on the observation that correct traces tend to exhibit larger confidence gain from early to late tokens than incorrect ones. Reviewers broadly agreed that this empirical phenomenon is clearly demonstrated, that the method is lightweight and easy to apply, and that the theoretical analysis provides useful intuition for why such a pattern may arise in GRPO-trained models. The paper is also generally well written and easy to follow.

I therefore recommend acceptance.